# Downregulation of extraembryonic tension controls body axis formation in avian embryos

Daniele Kunz[1,2], Anfu Wang[1], Chon U Chan[3], Robyn H. Pritchard[4,5], Wenyu Wang [5], Filomena Gallo[6], Charles R. Bradshaw [1], Elisa Terenzani [1], Karin H. Müller[6], Yan Yan Shery Huang [5] & Fengzhu Xiong [1,2] ✉

Embryonic tissues undergoing shape change draw mechanical input from extraembryonic substrates. In avian eggs, the early blastoderm disk is under the tension of the vitelline membrane (VM). Here we report that the chicken VM characteristically downregulates tension and stiffness to facilitate stage-specific embryo morphogenesis. Experimental relaxation of the VM early in development impairs blastoderm expansion, while maintaining VM tension in later stages resists the convergence of the posterior body causing stalled elongation, failure of neural tube closure, and axis rupture. Biochemical and structural analysis shows that VM weakening is associated with the reduction of outer-layer glycoprotein fibers, which is caused by an increasing albumen pH due to $CO_2$ release from the egg. Our results identify a previously unrecognized potential cause of body axis defects through mis-regulation of extraembryonic tissue tension.

One of the major goals of developmental biology is to understand how large-scale changes in tissue shape (morphogenesis) are regulated at the small-scale molecular level. To drive deformation of soft biological matter in space and time, molecular mechanisms need to generate tissue forces and regulate tissue mechanical properties. While emphasis has been placed on how developing tissues self-organize mechanics internally by controlling cell behaviors, the regulation of the mechanical environment in which the tissues develop is much less understood.

During the early development of many animals, the embryo proper is structurally supported by extraembryonic tissues, which are further attached to the components of the egg. These connections allow transmission of materials, signals and forces between the embryo and the egg environment. As embryonic tissues change shape, they may be assisted or resisted physically by the connected structures[1]. Measuring the mechanical dynamics of these structures and elucidating their regulatory mechanisms is therefore important

for our understanding of developmental defects and the engineering of embryoids and organoids.

Avian embryos offer an excellent model for studying the developmental role and regulation of tissue mechanics because of their large size and accessibility. Inside the egg, the early embryo is supported by a multi-layered protein structure enclosing the yolk, known as the vitelline membrane (VM). The layers of the VM mainly consist of networks of glycoproteins, which provide structural integrity and share homologies to some components of the zona pellucida in mammals[2–4]. In chicken embryos, the outer rim of the blastoderm stays attached to the VM during early development, allowing tension transmission between the VM and embryonic tissues[5,6]. The disc-shaped blastoderm expands outwards prior to and during the first day after laying (D0, 0–24 h) through extensive proliferation and cell movements at the blastoderm edge (epiboly, Fig. 1a)[5,7,8].

During the following two days (D1-D2, 24–72 h), the gastrula embryo converges prominently along the primitive streak and the

[1]Wellcome Trust / CRUK Gurdon Institute, University of Cambridge, Cambridge, UK. [2]Department of Physiology, Development and Neuroscience, University of Cambridge, Cambridge, UK. [3]Institute of Molecular and Cell Biology, A*STAR, Singapore, Singapore. [4]Department of Physics, University of Cambridge, Cambridge, UK. [5]Department of Engineering, University of Cambridge, Cambridge, UK. [6]Cambridge Advanced Imaging Centre, University of Cambridge, Cambridge, UK. ✉e-mail: fx220@cam.ac.uk

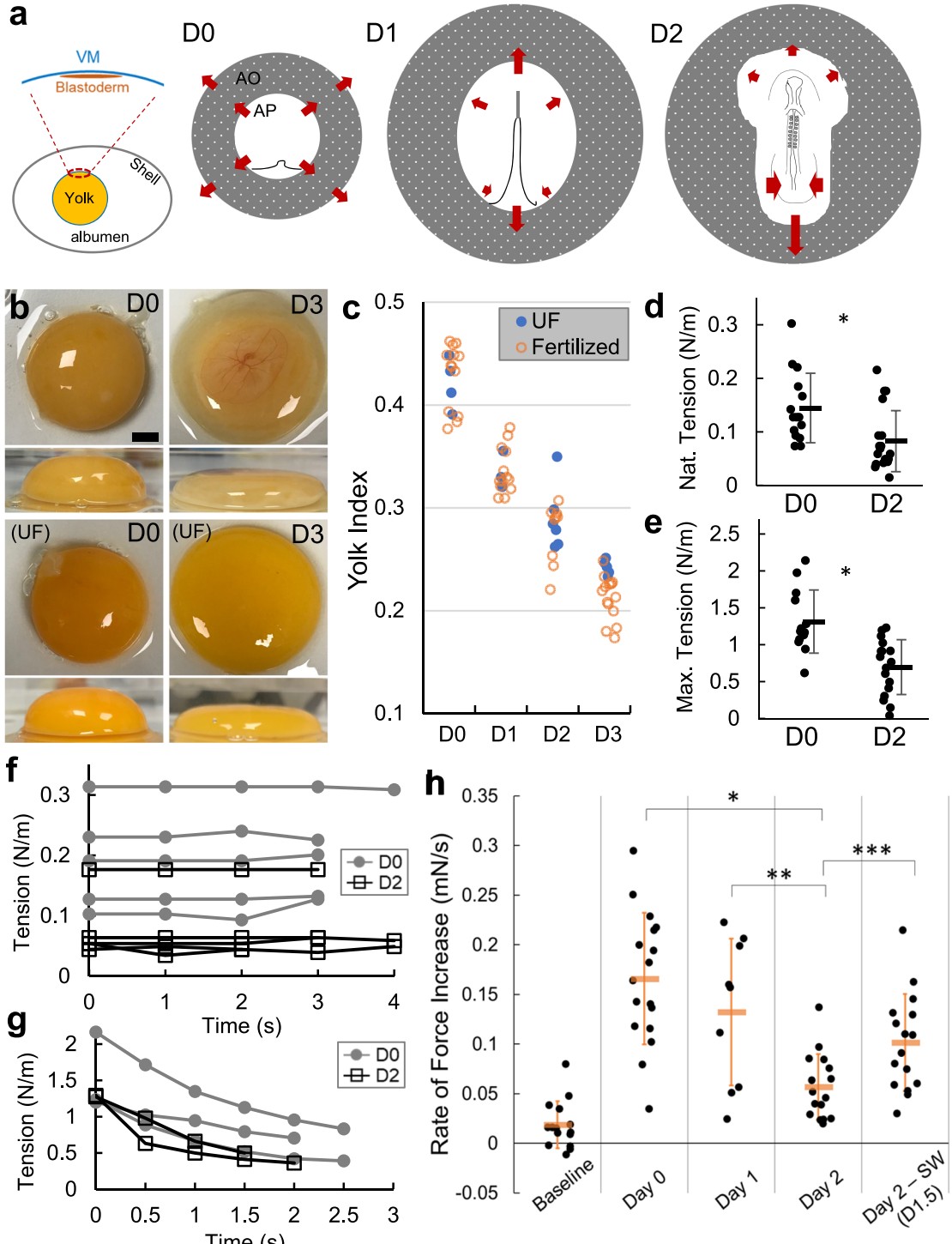

**Fig. 1 | VM tension and strength decrease during early stages of development.**
**a** Diagram of early embryo tissue flow. Left panel shows the blastoderm in relation to the vitelline membrane (VM) and the egg. Right panels illustrate the early shape changes of the blastoderm including area opaca (AO) and area pellucida (AP). Each illustration (anterior to the top) shows the configuration of the blastoderm at approximately the beginning of the labeled incubation day (D0, D1 and D2). D0 involves blastoderm expansion and extension of the primitive streak. D1 and D2 involve continued regression of the streak, tissue convergence and body axis elongation that progress in an anterior to posterior order. Arrows indicate direction of tissue flow. **b** Yolk deformation on a flat surface at different stages (representative samples). Top and side views. UF unfertilized. Embryo is visible in the D3 image through the red-colored vasculature on the top center of the yolk. Scale bar: 1 mm. **c** Yolk index (height/diameter) over time. There is no significant difference between UF ($n = 18$) and fertilized ($n = 50$) in D0-D2 ($p = 0.65$ [D0], 0.96 [D1], 0.42

[D2], and 0.008 [D3], 2 tailed $t$-tests). **d**, **e** Native and maximum tensions sustained by the VM. Bars indicate mean +/− SD. D0, $n = 15$; D2, $n = 19$. *$p = 0.006$ and $p = 2e-4$, 2 tailed t-tests. -20% of D2 measurements were on VMs taken from unfertilized eggs. **f**, **g** Short-term dynamics of native ($n = 10$) and maximum tension ($n = 5$). Each trace represents a single sample. Measurements are taken while the holder is static and the VM is intact. **h** The rate of force increase at a stable rate of pushing on the VM. Stiffer samples will result a faster increase. Baseline samples do not contain VMs. Day labels indicate the amount of incubation time before the VM samples are measured ($n = 15, 17, 9, 16, 16$). Texts following Day labels indicate perturbation type (after the dash) and perturbation start time (in "()"), respectively. For example, "SW (D1.5)" indicates filter sandwich (SW) perturbation started at D1.5. Same for subsequent Figures. Bars indicate mean + /− SD. *$p = 2e-6$, **$p = 0.005$, ***$p = 0.005$, 2 tailed $t$-tests.

forming body axis, reversing local tissue flow direction (Fig. 1a)[9–12]. Most ex ovo culture studies of the chicken embryo found that a stretched VM and its attachment to the blastoderm are necessary for proper long-term development[13–18]. In the absence of the VM, the blastoderm can only develop when it is supported on a gel substrate[19] (with posterior axis abnormalities), or when the tension is supplied by inflation through water movement in a "Cornish pasty" culture[20,21]. These observations suggest that the extraembryonic physical environment provided by the VM is critical for embryo morphogenesis in ovo.

In this study we show that the dynamics of VM mechanics play an essential role in normal development. First, we find that the tension and stiffness of the VM are downregulated during the first 48 h of incubation using mechanical measurements. Second, we use a combination of tension perturbation approaches to show that the early, strong VM is required for blastoderm expansion; while the later, weakened VM is required for the convergence and elongation of body axis tissues. Third, we show that the structural changes that underlie the mechanical changes of the VM result from the pH increase in the albumen. Swapping albumen, lowering pH and reducing $CO_2$ diffusion through the eggshell can biochemically retain a stronger VM, which causes similar body axis phenotypes to those from mechanically increased VM tension. Together, our results show that correct biochemical regulation of the physical properties of the extraembryonic environment is critical for normal embryogenesis.

## Results

### Vitelline membrane weakens during early development

To understand the mechanical dynamics of the VM, we first looked at its effect on the shape of the yolk which it encloses. The yolk index (height/diameter of a separated yolk on a flat surface) is a measure of egg quality in poultry science and reduces characteristically during incubation (Fig. 1b, c)[22]. The presence of a developing embryo does not alter these dynamics through D3, showing that the cause of change does not come from the embryo itself. Because yolks do not show significant changes of volume or mass during these stages[23] (Supplementary Fig. 1a), the VM which wraps the yolk maintains a stable surface area in ovo. When placed on a flat surface, the tension of the VM balances gravity (which drives the spread of the yolk) at mechanical equilibrium, producing a puddle-like shape with enlarged VM surface area. Therefore, the observed reduction in yolk index (and corresponding increase of VM surface area) is mainly caused by the VM becoming increasingly stretchable under a constant yolk weight. We modelled the yolk shape by relating the height of the yolk puddle to the VM tension (Supplementary Fig. 1b), giving VM tension in the order of magnitude of 0.1–1 N/m. The VM tension on D2 is predicted to be ~40% of that on D0 from the differences in puddle shape.

To directly measure VM tension, we extracted a piece of VM from the blastoderm area using windowed filter paper that sticks to the VM[15] and then sandwiched it with a second piece of filter paper to maintain its endogenous stretched state. The sample (kept wet with albumen during the measurements to maintain its in ovo chemical environment) is then connected to a weight on one end and vertically suspended on the other. Cutting the horizontal filter paper connections allows the weight to be supported by the VM alone, which would reduce the weight reading by a certain amount due to the existing/native tension on the VM (Supplementary Fig. 1c, d). By raising the holder, we could increase the tension on the VM until it reaches a breaking point (maximum tension is read before a sharp drop at rupture). We found that as incubation proceeds, the VM sustains a reduced level of both native and maximum tension, in both fertilized and unfertilized eggs (Fig. 1d, e). The maximum tension is consistent with previous studies using pipette aspiration and texture analyzer experiments on the yolk to evaluate VM tensile strength[24,25]. We also noted that the VM behaves elastically at native tension levels (tension is non-zero and remains stable over time, Fig. 1f) but relaxes quickly at

higher levels while the holder maintains the strain (Fig. 1g). While this ex ovo method likely introduces some variability during sample preparation, the average magnitude and difference between the VM tension at D0 and D2 are consistent with our model estimation (Supplementary Fig. 1b). These results show that VM is under tension which decreases from D0 to D2.

To assess the mechanical properties of the VM, we constructed a measurement device that includes a force sensor mounted on a moving stage, a scaffold to hold the VM samples and a side camera (Supplementary Fig. 1e, f, Supplementary Software 1). We used the probe to push perpendicularly on the center of the VM sample attached to filter paper, to stretch the VM while observing its deformation. This method provides a controlled and efficient way to assess VM mechanics as it is compatible with filter-paper-based ex ovo culture of the embryos[15]. The force detected by the probe increases as the VM becomes more stretched. We found that D0 and D1 VMs produce higher tension under the same deformation as compared to D2 VMs (Fig. 1h). While these deformations likely result in tensions beyond the native levels, the detected differences suggest a sharp reduction of VM stiffness between D0 and D2. Our mechanical model and measurements together show that VMs weaken characteristically during the early stages of embryonic development.

### VM mechanics affect development of the early embryo

To test the role of VM mechanical changes on embryo morphogenesis, we first extracted yolk in ovo at D0 to cause a reduction of pressure on the VM. The VMs showed visible wrinkles after yolk extraction indicating relaxation of tension. The eggs were then resealed and incubated for 6 h and up to D1.5, respectively, to assess the short-term and longer-term morphogenetic effects. Using GFP embryos[26], we found that the blastoderm sizes were smaller indicating delayed epiboly (Fig. 2a, b). Consistently, in yolk-extracted eggs incubated to D1.5, the area pellucida showed reduced expansion and consequently the body axis length (head to tail) was shorter by >60% (Fig. 2c–e). These phenotypes resemble explanted cases where the VM is manually ruffled to relax tension[13]. Control experiments were conducted in which the yolk was extracted and reinjected immediately. These embryos do not show expansion defects. These results confirm observations in previous studies[5,6] that the early VM needs to be under tension for proper blastoderm expansion.

Conversely, we tested the effect of preventing the observed reduction in tension between D1 and D2 by sandwiching the normal ex ovo culture[15] on D1.5 with an additional piece of windowed filter paper (36–40 h, Supplementary Fig. 2a). At this stage, the embryo is undergoing gastrulation and body axis formation. The sandwich manipulation limits the ability of the VM and extraembryonic tissues to move or stretch, resulting in an effectively stiffer tissue environment that is mechanically more similar to D1 rather than the actual stage of D2 (Fig. 1h). Under this condition, tissue tension is expected to rise as the large-scale convergence movements underway during these stages pull on the extraembryonic tissues and the VM. To confirm that the embryonic tissues are under a higher tension in sandwiched cultures, we performed surgical cuts through the endoderm and ectoderm lateral to the body axis. The wounds open wider and more persistently in sandwiched embryos after the initial cut but not in control embryos, indicating a higher level of tension in the tissues (Supplementary Fig. 2b, c).

Under enhanced tension, the sandwiched embryos start to show a delay in convergence and elongation of the paraxial mesoderm and the neural tube in the posterior body axis (Fig. 3a–e; Supplementary Movie 1–3). Prolonging this condition to ~6 h results in widening of the axis structures and cases of tissue rupturing (~30%, Fig. 3c; Table 1). The ruptures initiate in the posterior midline progenitor region or within the posterior paraxial mesoderm (Supplementary Movie 3), and then propagate into the neural midline or the paraxial mesoderm along the anterior-posterior axis. These rupture-initiating areas are

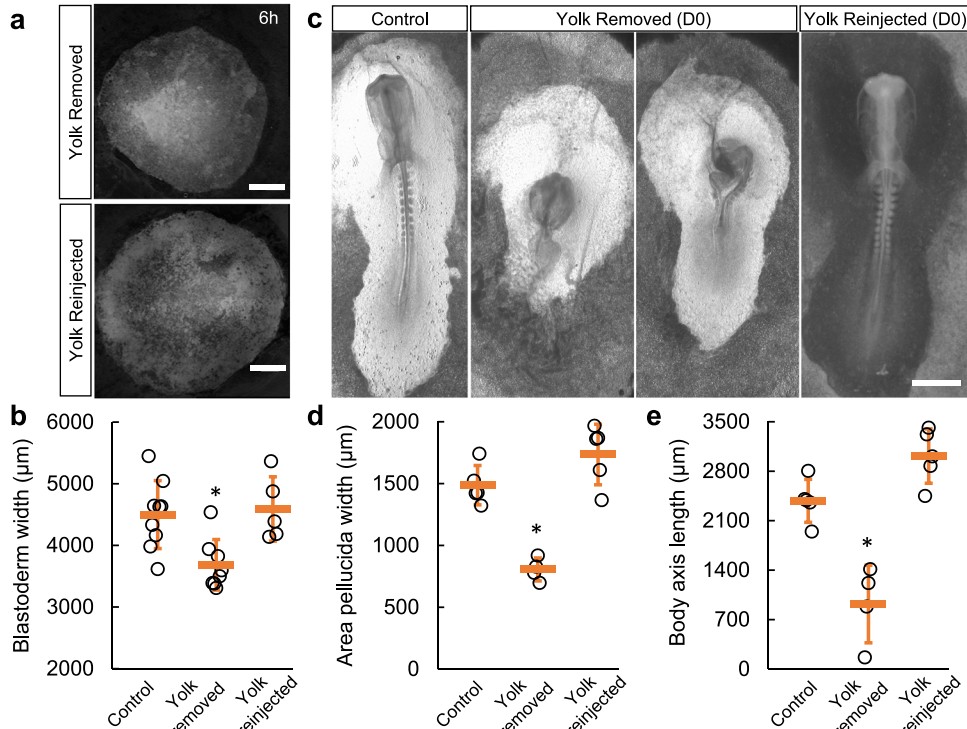

**Fig. 2 | D0 higher VM tension is required for blastoderm expansion.**
**a** Representative GFP+embryos following yolk extraction and reinjection (6 hrs).
2.5 ml yolk (-15% V/V) was removed by a syringe with a thin needle through an open
window on the eggshell. Wrinkles of VM could be observed after yolk extraction.
Eggs with no yolk-leaking were then resealed with tape and incubated. Whole
blastoderm images were taken with tile-scan from the dorsal side. Scale bar: 1 mm.
**b** Average width (diameter) of the blastoderm. Bars indicate mean +/− SD. *on the

yolk-removed group ($n = 8$) indicates $p = 0.004$ vs. control group ($n = 9$) and
$p = 0.005$ vs. reinjected group ($n = 5$), 2-tailed $t$-tests. **c** Representative embryo
phenotypes (D1.5) following yolk extraction and reinjection. Scale bar: 1 mm.
**d, e** Width of the area pellucida measured on the posterior end of the body axis and
body axis length (head to tail,) after yolk perturbations ($n = 5, 4, 5$, respectively).
Bars indicate mean+/−SD. * indicates $p = 3e-4$, $p = 0.001$ vs. both groups, 2-tailed
$t$-tests.

known to have high cell motility and a reduced level of cell-cell cou-
pling which may underlie their mechanical weakness under tension[11,27].
In contrast, in the anterior body, the somites and neural tubes show a
smaller degree of widening under prolonged sandwich culture and do
not initiate rupture. The somite formation rate was slower in the
sandwiched embryos.

A majority (-70%) of embryos under the sandwich culture do not
rupture but develop a wider and shorter body axis as compared to
controls (Supplementary Movie 1, 2; Fig. 3b, d, e). This suggests that
the mechanical changes in the VM and the tissues caused by this per-
turbation are not excessive and is consistent with the mechanical
measurements (Fig. 1h). Notably, the embryos exhibit a significant
delay of neural tube narrowing and, in some cases (-20%), even a
reversal of convergence (resulting in widening). Using confocal
microscopy, we found a much flatter, less folded posterior neural plate
in sandwiched embryos as compared to the controls (Fig. 3f, g). These
data show that a higher level of tissue tension impairs the ability of the
neural folds to meet and potentially prevents tube closure along the
dorsal midline. The increased tension is likely transmitted to the neural
epithelium via the non-neural ectoderm, which is known to play a
mechanical role during neural tube closure in different systems[28–32].
Indeed, some of the sandwiched embryos show a permanently open
posterior neural tube despite developing a normal tail fold (Fig. 3h),
resulting in a phenotype that resembles mammalian Neural Tube
Defects (NTDs) such as spina bifida[33]. To further examine the tissue
changes under increased tension, we performed scanning electron
microscopy on cross-sections of the body axis (Fig. 3i, j). Interestingly,
the neural tube and the notochord flatten as a whole while the non-
neural ectoderm appears to thin and separate from the presomitic
mesoderm (PSM), which appears to have lower cell density. These

observed changes occur gradually over an extended period of time
(-6 h), indicating that different tissues deform and change cellular
organization differently under tension.

While the sandwich method takes advantage of the ex ovo filter
cultures to provide a high-throughput way to manipulate tissue ten-
sion, allowing a detailed analysis of phenotypes, its effect on VM ten-
sion is indirect. To increase VM tension directly, we constructed a
double-ring culture device that pins down the VM over a liquid
chamber connected to an external syringe enabling pressure control
(Supplementary Fig. 2d, e)[17]. Embryos show strain when the VM is
inflated by media injection, consistent with tension transmission
between the VM and embryonic tissue[5,34]. Stepwise injection causes the
posterior neural folds to widen, and eventually the tissues rupture
after multiple rounds of inflation (Fig. 4a, b, 10/10). The rupturing
points under these conditions also initiate in the posterior midline and
paraxial mesoderm. These results mirror the sandwich experiments,
suggesting that a moderate increase of VM tension resists convergence
while a large increase causes tissue rupture.

The double-ring method works on short timescales and intro-
duces increased pressure on the tissues to increase tension. To dis-
tinguish between tension versus pressure as direct causes of the
phenotypes, we designed another approach – equiaxial stretcher–to
alter VM tension. Similar to a camera shutter, the stretcher uses rota-
tional movement of the blades to generate center to periphery
movement (Supplementary Fig. 2f), and when attached to the VM via
filter paper discs, modulates VM tension by changing VM strain
(Fig. 4c). We found that embryos loaded on the stretcher show global
tissue expansion (Fig. 4d). This confirms that tension on the VM is felt
throughout the embryo, as only the VM is in contact with the stretcher
device, unlike in the sandwich experiments where extraembryonic

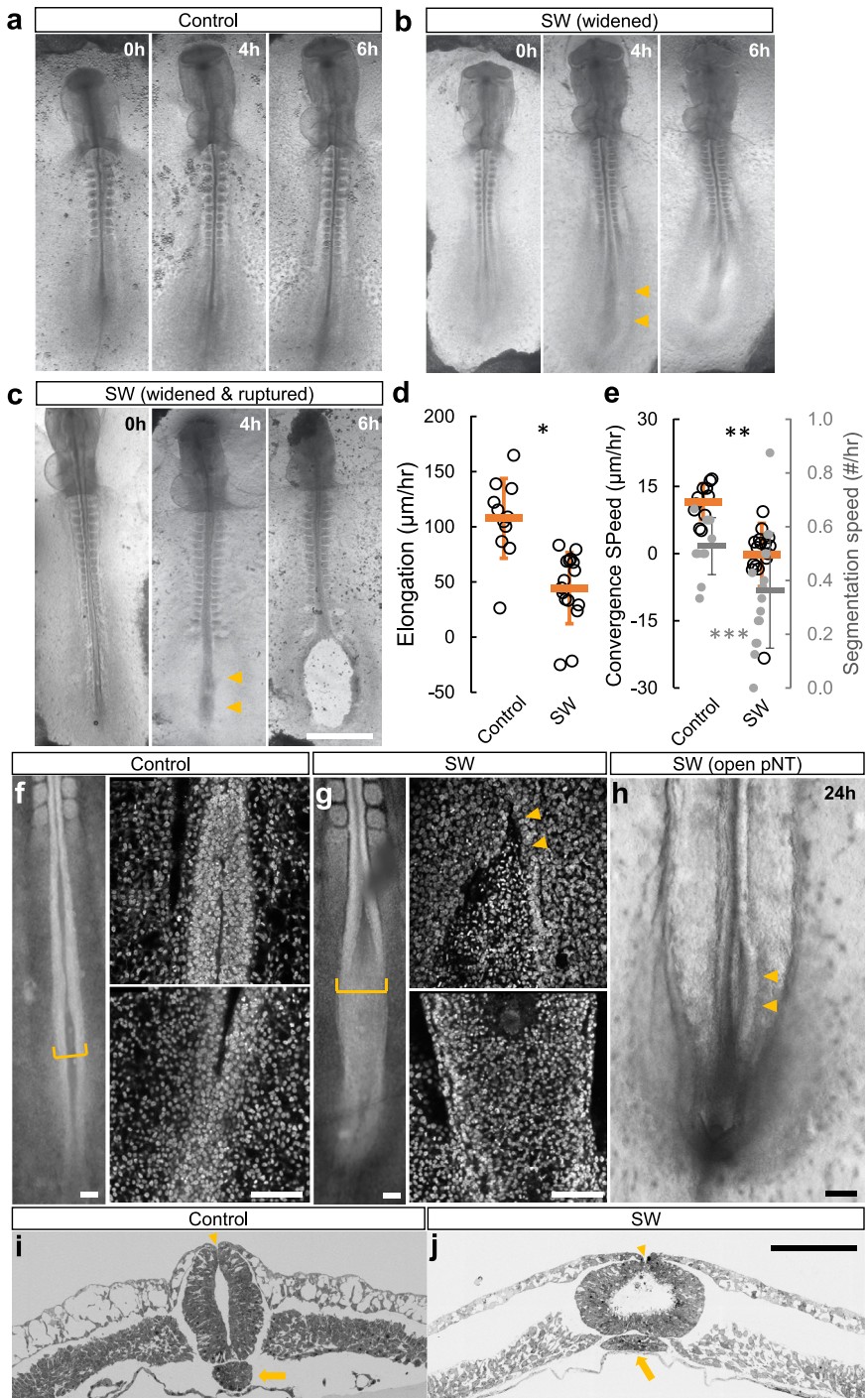

**Fig. 3 | Maintaining higher VM tension in D1-D2 impairs body axis morphogenesis. a–c** Embryo phenotypes (D1.5-D2) following SW. Time stamps indicate hours after the addition of the second filter. Arrowheads indicate the widening posterior body axis. A fraction of SW embryos ruptured under 6 h ($n = 11/37$). No rupture was observed in controls ($n = 0/20$). Scale bar: 1 mm. **d, e** Elongation and convergence speeds measured 6–8 h after SW in embryos that did not rupture ($n = 11$, control; $n = 16$, SW). Some embryos showed a negative convergence speed i.e. widening and/or a negative elongation speed i.e. shortening. The segmentation speed in SW group is also slower than the control group (grey points in **e**). Bars indicate mean +/− SD. *$p = 7e{-}5$, **$p = 3e{-}5$, ***$p = 0.03$, 2 tailed *t*-tests. **f, g** Confocal images of the posterior axis comparing Control and SW (nuclear DAPI signal, 6hrs post SW). In the image showing the control embryo the posterior neural tube appeared narrow and mostly closed, while in the SW embryo the posterior neural tube was open and wider. The higher magnification views show the neural tube near the forming somites (top) and the posterior end (bottom), respectively. Scale bars: 100 μm. See also Supplementary Movies 1-2. **h** Open posterior neural tube (pNT) phenotype ($n = 3/10$, arrowheads) in extended SW culture at 24 h post SW (D2.5). Scale bar: 100 μm. **i, j** Scanning electron microscope (SEM) cross-sections of the posterior body axis at the level of the closing neural tube (arrowheads point to the dorsal junctions of the closing neural folds). 6 h post SW. Arrows point to the notochord. Scale bar: 100 μm.

**Table 1 | Summary table of qualitative phenotypes**

| Approach | Experiment | Expected effect on Tension/VM (no effect = natural downregulation) | Duration of treatment | Body axis phenotypes (D1.5-D2) | | | Total |
|---|---|---|---|---|---|---|---|
| | | | | VM rupture | No body axis | Axis rupture | |
| Yolk extraction | Extracted | reduce | 6 h or up to D1.5 | | 2 | 0 | 14 |
| | Control group (Yolk re-injection) | no effect | 6 h or up to D1.5 | | | 0 | 10 |
| | Control group (Unperturbed) | no effect | 6 h or up to D1.5 | | | 0 | 14 |
| Filter sandwich | Sandwiched | maintain/increase | 6 h from D1.5 | | | 11 | 37 |
| | Control group (Single filter) | no effect | 6 h from D1.5 | | | 0 | 20 |
| Fluid inflation | Inflated | increase | 2–10 min | | | 10 | 10 |
| Equiaxial stretcher | Loaded on stretcher | increase | 6 h from D1.5 | | | 3 | 15 |
| | Control group (not loaded) | no effect | 6 h from D1.5 | | | 0 | 10 |
| Acid injection | Injected group | maintain | D0 to D2 | | 8 | | 8 |
| Heterochronic Albumen transfer | D0- > D0 | no effect | until D2 | 0 | 1 | 0 | 4 |
| | D2- > D0 | reduce/weaken | until D2 | 3 | 2 | | 5 |
| | D1- > D1 | no effect | until D2 | 0 | 0 | 0 | 8 |
| | D0- > D1 | maintain | until D2 | 0 | 0 | 0 | 8 |
| pH plate culture | Acidic plate (pH = 6.5) | maintain | 6 h from D1.5 | | | 0 | 19 |
| | Control group (pH = 9.0) | no effect | 6 h from D1.5 | | | 0 | 18 |
| Oil treatment | 1/2 Oiled D0 | maintain | D0 to D2 | 0 | 1 | 3 | 21 |
| | Oiled D1 | maintain | D1 to D2 | 0 | 1 | 4 | 20 |
| | Oiled D2 | maintain | D0 to D2 | 0 | 3 | 10 | 23 |
| | Control group (no oil) | no effect | D0 to D2 | 0 | 0 | 0 | 7 |

(white spaces indicate count not applicable or not recorded).

tissues are also directly affected in addition to the VM. In particular, the posterior body axis tissues (in the example in Fig. 4d', the pNT) shows widening under increased tension. The strain introduced by the stretcher is stable and allows culturing of the embryo (along with the stretcher) over hours, matching the timescale of morphogenesis. We found that the pNTs in embryos mounted on the stretcher show much slower convergence (Fig. 4e). Further increasing the rotation of the stretcher blades also leads to tissue ruptures. These phenotypes are similar to the sandwich and inflation experiments and further demonstrate that maintaining VM tension at a higher level is sufficient to disrupt axis convergence in the absence of direct perturbations on the extraembryonic tissues (sandwich method) or pressure (double ring method).

While each approach of tension manipulation has its caveats, they are consistent in producing axis-widening phenotypes that show significant delay in convergence and in some cases ruptures as tension further increases (Table 1). These results are consistent with the hypothesis that a lowered VM tension is required for normal posterior body morphogenesis between D1 and D2. Our mechanical approaches take advantage of the ease of access to the VM in the ex ovo embryo culture. To find methods that prevent tension decrease in ovo to further test our hypothesis, a biochemical understanding of VM weakening is required.

**Albumen pH increase causes VM protein loss and mechanical weakening**

To identify the mechanism of change in VM mechanics, we performed electron microscopy to examine the structural changes in the VM during egg incubation. We found a major reduction of protein density in the outer layer of D2 samples as compared to D0 (Fig. 5a, Supplemental Fig. 3a). This is consistent with an overall loss of proteins including glycoproteins on the VM (Supplementary Fig. 3b). We stained for glycoproteins directly on the VM and observed a significant

reduction of fiber density in D2 VMs as compared to D0 (Fig. 5b, i, Supplemental Fig. 3c). Previous studies have shown that the most drastic chemical change in the egg environment during these stages is an increase in the pH of the albumen from ~7.5 upon egg laying (when the albumen has a similar $CO_2$ content to the hen's body) to 9-10 on D2[35,36], so we tested the effect of pH on isolated VMs. Indeed, treating D0 VMs with a pH 9.3 buffer greatly reduces glycoproteins within 12 h while the fibres are more preserved under pH 7.5 (Fig. 5c, i, Supplemental Fig. 3c). The mechanical weakening of VMs in buffer also progresses faster at the incubation temperature (Supplementary Fig. 3d). An increase in pH is known to weaken the interaction strengths between glycoproteins, which reduces the viscosity of the thick albumen and potentially also affects the protein composition of the adjacent VM outer layer[37]. We repeated the biochemical assay testing the interaction between ovomucin (a VM glycoprotein) and lysozyme (another major VM component) and confirmed that the solubility of the mixture increases with pH (Fig. 5d). These results show that pH controls VM structural integrity which may underlie its mechanical changes during incubation. To test if this biochemical mechanism is reversible, we performed experiments where the extracted VMs are subjected to co-incubation with the albumen from a different date (Fig. 5i, Supplemental Fig. 3c). We found that incubated albumen in D2 could decrease the fibre density of D0 VMs but D0 albumen could not reverse the loss of fibres in D1 and D2 VMs, suggesting that the structural decay of the VM is not reversible. Another possible factor modulating VM decay is the tension itself. We tested this idea by measuring VM fibre density in the SW samples. We found that SW condition does not cause a density change after 6 h (Fig. 5i), suggesting that higher tension does not promote or inhibit VM structural change.

Direct alteration of the albumen pH in ovo by adding acids proved challenging, as the albumen becomes murky (probably as a result of albumen protein denaturation) and embryos do not develop. As an alternative, we performed heterochronic albumen transfer. Eggs

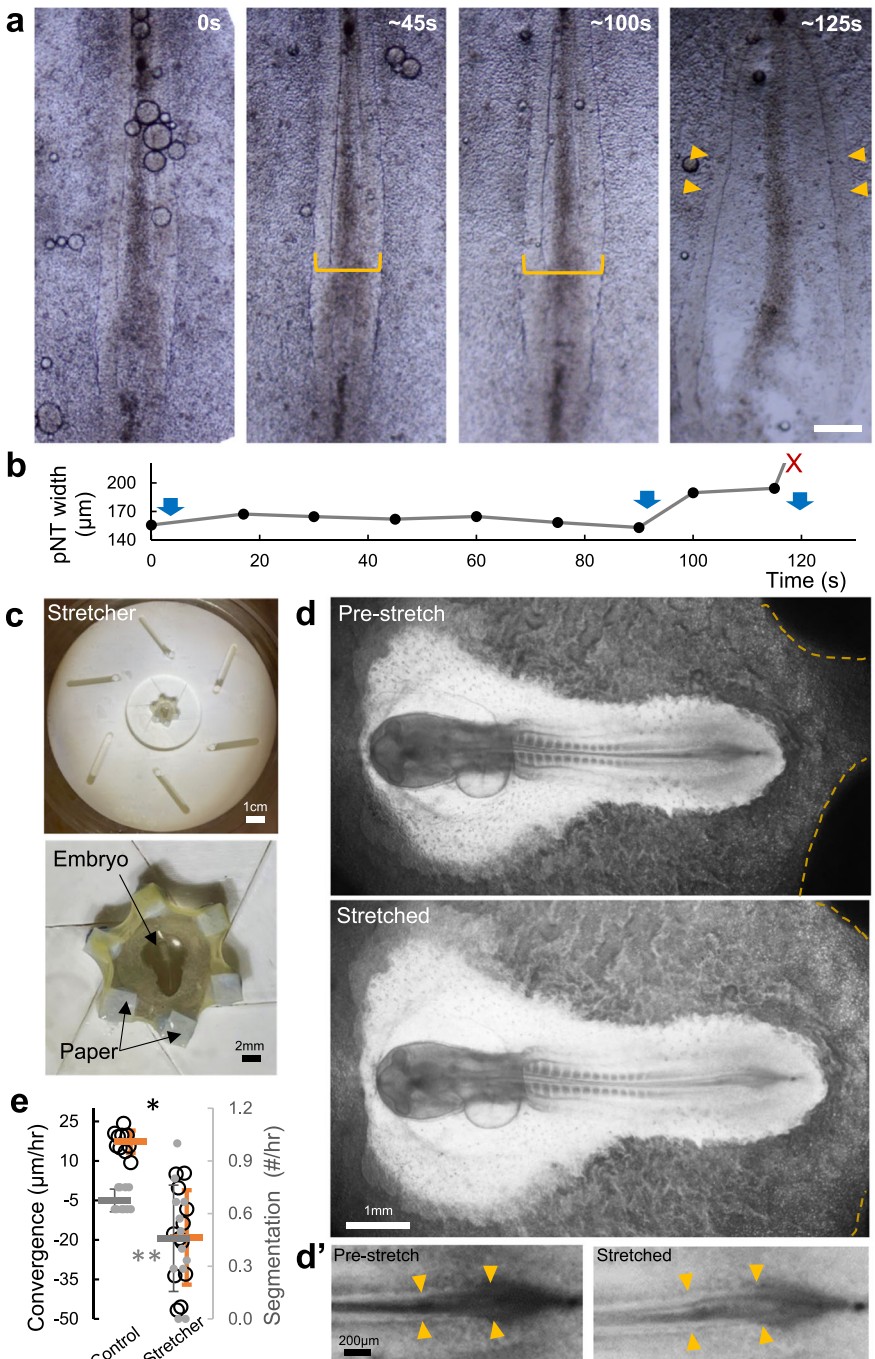

**Fig. 4 | Increasing tension via inflation and stretching causes axis widening.**
**a** Embryo phenotypes following inflation culture. Time stamps indicate seconds after the start of live recording. Yellow brackets measure pNT widths. Arrowheads point to the NT tissue after rupture. Posterior body axis widening and/or rupture was observed in all samples tested ($n = 10/10$). Scale bar: 100 μm. **b** Width of the pNT in inflation culture. Blue arrows indicate times of additional injections of medium to increase inflation and tension. Widening and rupture (red X) follow the injections. **c** Rotational stretcher device and example of a mounted embryo. **d** Short-term stretching using the stretcher on the VM causes global tissue expansion. The dashed lines mark the contact areas between two of the movable studs and the VM. Under stretching, the embryonic tissues, AP and AO can be seen to enlarge. **d'** shows a magnified view of the pNT region where the neural folds can be seen to widen. **e.** Longer-term convergence speeds measured on the pNT for the ~3 h window after the embryos were mounted on the stretcher ($n = 12$). The controls ($n = 9$) were not mounted. The segmentation speed (grey data points) of the embryos on the stretcher becomes variable and slightly reduced on average. Bars indicate mean +/− SD. *$p = 2e-5$, **$p = 0.04$, 2 tailed $t$-test.

incubated to D1 had their albumen removed and replaced with fresh albumen (lower pH) from D0 eggs. These embryos showed a shorter body axis (Fig. 6a), consistent with phenotypes of increased VM tension ex ovo. The axis lengths in the control transfer (D1 to D1) are indistinguishable from non-operated embryos. For an internal control of body axis length, we compared somite numbers between the experimental groups, which confirmed that embryos were at similar developmental stages (Fig. 6a). We also performed the reverse experiment where D2 albumen was moved to D0 eggs. Interestingly, these eggs show an extremely fragile VM as some yolks were broken already when the eggs were opened and others broke upon opening, preventing embryo retrieval (Table 1). This result is however consistent

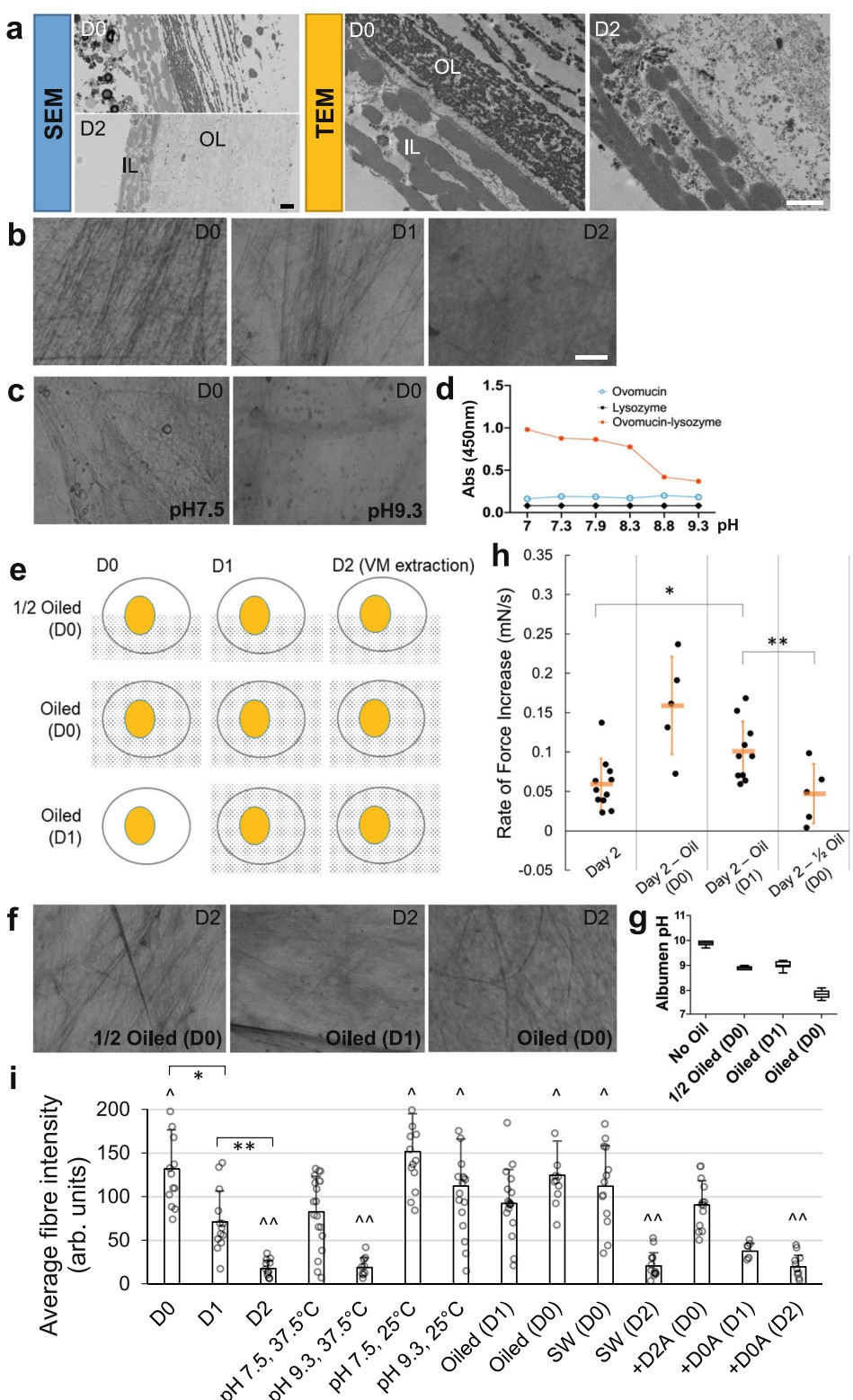

with the hypothesis that a higher pH albumen environment (D2) transferred to D0 eggs accelerates the decay of VM. We also conducted ex ovo experiments by culturing the embryos on acidic (pH 6.5) plates and compared these to the normal culture plates (pH 9.0). Embryos on the acidic plates showed slower body axis elongation and convergence speeds (Fig. 6b, c). While these results are consistent with tension phenotypes and support a link between pH-controlled VM mechanical properties and embryo development, the disruptive procedures of heterochronic albumen transfer introduce a high degree of variability

in phenotypes, and acidic plates may cause posterior axis phenotypes via pathways[38] other than preventing VM weakening.

To further reduce the invasiveness of pH manipulation, we used mineral oil to coat the eggshell (Fig. 5e), which blocks the release of $CO_2$ from air pores that is responsible for the pH increase[35]. This perturbation indeed stalled both the albumen pH increase and glycoprotein fiber reduction (Fig. 5f, g, i). In particular, eggs fully coated in oil at D0 maintain a pH around 7.5 when measured at D2, similar to freshly laid eggs at D0 despite 48 h of incubation. Eggs that were half

**Fig. 5 | Biochemical changes in the egg environment control VM mechanics.**
**a** Scanning (SEM) and Transmission (TEM) electron microscope cross-sections of the VM (representative samples). OL outer layer, IL inner layer. Scale bar, 1 µm. **b, c, f** Glycoprotein fibers on the VM stained by the PAS (Periodic acid–Schiff) method. Top right label shows the incubation time when the VM was extracted. Bottom right label shows perturbations. Oil treatments started from the Day indicated in "()". pH buffer treatments lasted 12 h. See Supplementary Fig. 3c for additional samples. Scale bar: 10 µm. **d** Interaction assay between Ovomucin and Lysozyme. Diluted preparations of two components fogged the solution when mixed. The interaction weakened with increasing pH. **e** Diagram of oil coating experiments. Dotted patterns indicate oil coating. **g** Albumen pH after oil treatments ($n = 8$ for each group). The whiskers mark the minimum and maximum measurements. The boxes are 25% to 75%. The lines are the median value. **h** Stiffness measurements of VMs from oiled eggs. *$p = 0.004$, **$p = 0.022$, 2 tailed $t$-tests. $n = 11, 5, 10, 5$, respectively. The method and experimental group labeling in the panel follow those of Fig. 1h. **i** Fibre density of PAS stained VMs. $n = 13, 13, 11, 19, 9, 14, 18, 17, 11, 13, 12, 13, 7, 11$ random areas were measured in each group of images, respectively. Bars are the mean value and error bars are SD. ^ indicates no group is significantly different from other groups including the D0 control ($p = 0.14$), ^^ indicates no group is significantly different from other groups including the D2 control ($p = 0.96$), One-way ANOVA tests; *$p = 0.0008$ and **$p = 9e-5$, 2 tailed $t$-tests. SW, filter sandwich for 6 h; +D2A and +D0A, incubated in D2 and D0 albumen for 24 h.

oiled at D0 and eggs fully coated at D1 both showed an intermediate pH at D2, higher than D0 but lower than control eggs. Importantly, we found that the VMs from fully oiled D0 and D1 eggs retain their respective stiffness (Fig. 5h), indicating that VM weakening is stalled by oil treatment. Strikingly, the embryos in the oiled eggs fully recapitulate the spectrum of posterior body axis widening and rupture phenotypes of those observed in the ex ovo sandwich, inflation, stretching, pH plate and the in ovo albumen transfer experiments (Fig. 6d–f). Consistent with maintaining the stiffest VM and highest tension, the eggs fully oiled from D0 produced the most ruptured embryos (Fig. 6e, Table 1). On the other hand, the embryos that developed an intact body axis in these eggs do not show other major defects except the body axis widening phenotypes, suggesting that our oil treatment is not causing systemic failure of development due to the blocking of gas exchange. This is consistent with the low gas exchange requirement of embryos in these early stages[39]. Together, these data support the hypothesis that an increase in albumen pH mediated by $CO_2$ escape causes VM structural decay during incubation, leading to the mechanical weakening of the VM. This process is necessary for the normal development of the posterior embryonic body, linking the biochemical environment of the egg to the morphogenesis of the embryo via mechanics.

## Discussion

Tissue morphogenesis does not occur in isolation, but in connection to other tissues and the extraembryonic environment. To some extent, tissues can regulate force production against changes in external resistance and promote robust deformation. However, such adaptation has its limits[40]. Our results show that, during the early stages of chicken development, tissue tension is regulated by biochemical modifications of the VM, assisting the tissues to achieve the correct size and shape. Interestingly, we found very similar embryonic phenotypes under both physical manipulation of tension and biochemical manipulation of the egg environment. The dynamics of changing tension are likely conserved across the avian phylogeny, as similar yolk shape changes are seen in other avian species[41,42]. The strong early VM allows extraembryonic cells to generate sufficient tension at the blastoderm edge to drive tissue expansion[6], and the weak late VM allows body axis tissues to converge to the midline against low resistance. The changes of $CO_2$ and albumen pH are triggered by incubation, thereby ensuring that the timing of VM weakening aligns with the developmental progress of the embryo. This provides an example of how tissue mechanics can be biochemically regulated for correct development.

In addition to the biochemical modification of the VM, water influx from the albumen to the yolk may contribute to tension regulation[43] as such influx may increase the volume of the yolk resulting in increased pressure on the VM leading to higher tension. The capacity of tissue water transport is seen in chicken embryos in the Cornish pasty culture condition without the VM[21] and may play a conserved role in size (and probably tension) control in mammalian early embryos[44]. However, in

the D0-D2 early-stage chicken embryos studied here, yolk size increase due to water transport is minimal[23] and does not appear to compensate for the tension reduction due to VM weakening. What the tension dynamics are like in different ex ovo culture methods remains to be understood. In the classical New's method[18], the embryos are mounted ventral-side up on a watch glass containing a pool of albumen and glass rings hold the VM under tension. This approach essentially maintains the VM-albumen local environment and likely allows the VM to relax tension during incubation in a similar manner to when in the egg. The filter paper dish cultures[15] used in this study and by others have a regular pH around 9, mimicking the albumen of D1-2 eggs that have lost the high body level $CO_2$, and are therefore an appropriate environment for VM weakening. It is worth noting that submerged sandwich cultures work very well with no tension phenotypes[16,45], this might be explained by our observation that VM weakening happens quickly when it is soaked in water-based buffers, therefore the sandwiches can no longer cause increased tension. Furthermore, sandwich experiments in late D2 embryos in this study do not show major body axis defects, suggesting that VM tension may no longer be important for the posterior body axis after the convergence stages. It should be noted that in ovo, the blastoderm edge cells (now far away from the body axis after D2) continue to move on the VM and expand the tissue area which would likely continue to maintain/require some level of tension between the VM[34]. Another interesting culture condition that is VM-free is the soft gel culture originally reported by Spratt[19]. In these experiments, the explanted blastoderms can spread and form a largely normal body axis with a shorter and smaller posterior. How the tissues re-establish tension on the gel substrate remains unclear as the tissue-substrate contact area is presumably expanded from just the blastoderm edge in the case of the VM. It is possible that more cells in addition to the edge cells participate in contact and contraction on the gel. These ideas can be tested by live imaging and traction force microscopy during blastoderm expansion in future.

How global mechanical changes in the VM interact with local stress anisotropy generated by patterned tissue behaviors[12] remains to be characterized. As development proceeds in the early stages (D0), the tissue behaviors of the blastoderm become non-uniform. While the area opaca continues to expand outwards, the area pellucida starts to become patterned (e.g., appearance of the posterior marginal zone and then the primitive streak) and shows organized movements[10] (Fig. 1a), which likely underpins tension anisotropy[46]. Whether the spatial pattern of mechanical properties of the VM[47] contributes to this process remains unknown. Our structural and mechanical analysis of the VM samples taken around the blastoderm reported here has not revealed spatial heterogeneity, suggesting that the dynamics of VM tension may be uniform around the embryo. Therefore, patterned regulation of tension within the blastoderm will be essential for morphogenetic movements other than the expansion while VM tension provides a global baseline. A recent study shows that the area opaca serves an induction role to the marginal zone[48]. It is tempting to speculate that these interactions may also alter the tension in the posterior area opaca and area pellucida, such as

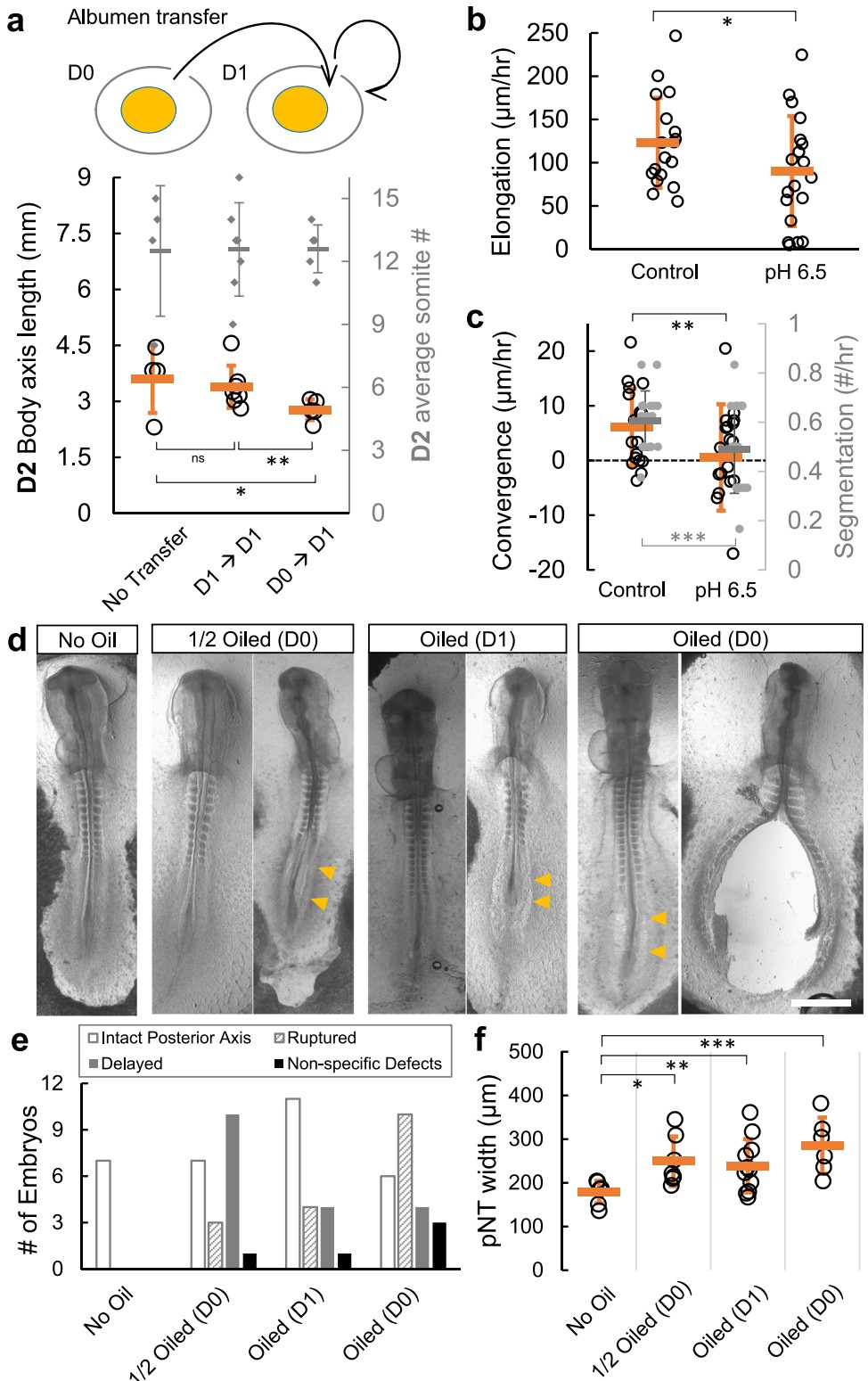

**Fig. 6 | Preventing VM weakening by pH manipulation causes posterior body axis defects. a** Heterochronic albumen transfer including D1 → D1 (control) and D0 → D1 fresh albumen into D1 eggs (test). Embryos were extracted at D2 and body axis length (from the second somite) was compared. # of somite pairs was counted as an internal control of developmental stage. The test group (*n* = 5) shows significantly (**p* = 0.046 vs. control, *n* = 4; ***p* = 0.025 vs. homochronic transfer, *n* = 7, 2 tailed *t*-tests) shorter axis at similar somite stages. Bars indicate mean +/− SD. **b**, **c** Elongation and Convergence speeds after 6hrs of incubation on acidic (pH 6.5) plates (*n* = 19 vs. control group, *n* = 18). Some embryos showed a negative convergence speed i.e. widening. The segmentation speed of the embryos on the acidic plates is also slightly reduced. Bars indicate mean +/− SD. **p* = 0.043, ***p* = 0.024,

***p* = 0.03, *t*-tests. **d** Representative embryo phenotypes following oil treatments. The embryos were extracted and imaged on D2. Images shown represent the variability of stages and phenotypes in oil treated embryos, summarized in (**e**). Arrowheads indicate widened posterior body axis tissues. Scale bar: 1 mm. **e** Summary of phenotypes following oil treatments. Aside from the ruptured embryos, those with 7–12 pairs of somites are categorized as intact posterior axis, embryos with fewer than 7 pairs of somites are categorized as delayed. Embryos with no discernable axis are categorized as non-specific defects. **f** Width of the pNT in embryos with intact posterior axis. Bars indicate mean +/− SD. No oil group, *n* = 7; **p* = 0.011, *n* = 7; ***p* = 0.026, *n* = 11; ****p* = 0.002, *n* = 6; 2 tailed *t*-tests.

reducing it to facilitate primitive streak formation. A spatial-temporal tension map at high resolution using novel access tools and sensors[49] for the VM and the blastoderm will help test these ideas in future. In later stages (D1-D2), the convergence of the body axis tissues is of particular interest given that it is an inward tissue flow and is the most strongly affected process under high tissue tension in our experiments. Convergent movement and folding of the neural plate may involve the non-neural ectoderm generating a pushing force at the tissue transition that aids neural fold formation[50]. One possible explanation for the failure of neural tube closure in our experiments is that increased tension on the VM limits the movement of these ectodermal cells, reducing the bending of the epithelial sheet that is important for proper folding. Alternatively, the non-neural ectoderm maintained under a higher level of tension could resist the contractile forces generated by the neural plate cells through apical constriction and interkinetic nuclear migration[51]. This would reduce tissue deformation while increasing the load on the cell junctions across the epithelial sheet, making it more likely for tissue ruptures to occur. In either scenario, the requirement of proper tension for tissue convergence raises possible mechanisms with which the neural plate and the non-neural ectoderm actively regulate tissue tension in response to environmental changes. Such mechanisms may include junctional remodeling and cell divisions (for example, cells orient the division axis to help relax tension[52]), which can be tested in future studies using our platform. The increased tension also has an effect on the mesenchymal tissue of the posterior PSM that is different from the widening of the epithelial tissues such as the neural tube and notochord. In this case the PSM appears enlarged with a decreased cell density. These changes could underlie the slowing of body axis elongation as PSM cell density is known to regulate elongation through cell motility and/or PSM mechanical properties[11,53,54]. In the anterior body axis, we observed relatively fewer changes of tissue morphology likely due to the tissues (e.g., somites, anterior neural tube) being stiffer. It is worth noting that in tension-increased perturbations a lower rate of somitogenesis is observed. This suggests that tension can interfere with the somite formation process, which is consistent with previous studies where somite deformation and fission were observed when the embryo was subjected to a much higher strain[55,56]. The observed changes to somitogenesis and elongation are unlikely to be associated but as separate responses to the globally applied tension perturbations.

The widened/open posterior neural tube in the more mildly perturbed embryos in our study mirrors certain neural tube defects that occur in ~1/1000 human newborns[33,57]. Our results may suggest a novel potential cause of neural tube defects through extraembryonic tension mis-regulation. Human embryos at the comparable stages have a similar disc geometry[58] and may be under tension through connections with the trophectoderm. The tension dynamics and regulatory mechanisms for the human blastoderm remain unknown and can potentially be studied using in vitro cultures or embryoids derived from stem cells[30,59,60]. Our results and tools in the avian model enable further studies of genetic and cellular changes under tissue tension, which will shed light on the causes of body axis defects in this critical stage of human development.

## Methods

This study does not require an animal protocol for the embryonic stages studied under the UK Animals (Scientific Procedures) Act 1986 (incubation of chicken eggs under two weeks).

### Avian eggs and incubation

Wild type fertilized chicken (*gallus gallus*) eggs were supplied by MedEgg Inc. Unfertilized eggs of similar sizes were obtained at Tesco. Transgenic GFP eggs were obtained from the National Avian Research Facility (NARF) at University of Edinburgh. Eggs were kept in a monitored 15 °C fridge for storage and 37.5 °C ~45% humidity egg incubators (Brinsea) for incubation. The eggs were incubated within 2 weeks of receipt. To measure the yolk shape, eggs were opened into a petri dish and both thin and thick albumen was removed carefully with a Pasteur pipette. A top and a side photo of the yolk was taken within 1 minute of albumen clearing, and within 5 minutes of egg opening. The yolk puddle shape reached stability within seconds.

### Light microscopy

Snapshots of the embryos were taken at different time points with a stereomicroscope (MZ10F, Leica). For timelapse imaging, cultured embryos were transferred to a 35 mm glass bottom dish (VWR) with a thin layer of culture medium (200 µl evenly spread on the glass bottom). Live imaging was performed using a Zeiss Axio Observer 7 microscope equipped with a motorized and heated stage using a 5x objective. Tiles were used to cover the whole embryo and stitched in Zen software to produce movies. For confocal imaging, embryos were first incubated for 6 h (control or after filter sandwich placement) and then fixed in 4% paraformaldehyde (PFA) at room temperature for 30 min. The nuclei were stained using DAPI (Sigma) 1/2000 for 10 min, followed by 2 washes in PBS. Embryos were then mounted on glass bottom dishes. The posterior body axis area of the samples was imaged using a laser scanning confocal microscope with a 40X objective (TCS SP5; Leica).

### Scale measurement of VM tension

Incubated eggs were opened into petri dishes and the top center of the yolk area was cleaned gently to remove the albumen. A strip of filter paper (Whatman) with two identically cut windows of 2 cm-wide was folded in the middle to create a loop for a holding pipette mounted on a manual stage. The VM was first attached to one side of the filter paper and cut from the yolk. Then the other side was folded over to fix the VM piece in place. A clip was attached to the open end of the fold to stabilize the mounting as well as serving as the weight on the sample. The whole assembly was then placed inside a digital balance scale (Fisher Scientific). The window has two small triangular cuts in the middle to allow easy release by cutting of the remaining filter paper supporting the weight. This also minimizes VM side deformation caused by the retraction at the cutting site due to tension. After cutting the VM suspends the weight alone. The scale reading of the weight reduces as the holder was raised in a few seconds until the VM breaks, when the reading returned to the maximum. The tension was measured as the force difference over the width of the VM. Specifically, $\gamma$(native) = [$m$(after_cut)-$m$(detached)]$g/L$, where $m$ is the weight measurement, $g$ is the gravitational acceleration 9.8 m/s$^2$, $L$ is the width of the VM samples (2 cm in the reported experiments). Similarly, $\gamma$(max) = [$m$(breaking)-$m$(detached)]$g/L$. See also Supplementary Fig. 1c, d.

### Mechanical probe system

A home-built system was used to measure the mechanical properties of the VM. The system consists of a manual stage (Drummond) with a home-made sample holder, a self-illuminating miniature microscope camera (Kern), and a force sensor (UF1 Isometric, LCMSystems) on a motorized linear translation stage (Thorlabs) connected to a data acquisition system (DAQ, National instruments). See also Supplementary Fig. 1e, f. The holder uses screws and custom-printed panels to fix the VM samples mounted on filter papers in place, allowing the system to accept samples prepared according to the embryo culture protocol (see the section below). The sensor data and the stage are controlled by a custom program created in LabView (National Instruments). During a measurement run, the VM sample was first mounted on the holder and the holder was then fixed on the manual stage which allows fine positioning of the VM sample to align perpendicularly to the force sensor. The sensor then started logging the measurements. Next the sensor moved towards the VM sample at a constant speed of 0.1 mm/s

while the approach, contact and VM deformation were recorded from the side by the camera. The pushing phase lasted 40 s with contact of VM generally occurring between 5 s and 15 s. After holding at maximum deformation for 10–15 s, the probe was retracted at the same speed.

## Embryo dish culture and filter sandwich

Embryos were staged following the Hamburger and Hamilton (HH) table[9] and were extracted for ex ovo culture using the Early Chick (EC) culture protocol[15]. Sex of the embryos is not a relevant variable at the early stages studied and was not assessed. For D1.5 controls, the embryos are mostly between stages HH9 and HH11 (approximately 8-somite to 12-somite). To prepare the culture medium, the following formula was used: First, for 100 ml of culture media, prepare Part A: 50 ml Albumen (beaten for 15 min) then supplemented with 0.8 ml 20% Glucose and Part B: 0.3 g BactoAgar (Sigma) dissolved in 50 ml water in a microwave then supplemented with 1.23 ml 5 M NaCl. Next, equilibrate part A and part B separately in a 55 °C water bath. Finally, mix thoroughly and pipette 2 ml each to 3.5 cm petri dishes before gelation. To make pH 6.5 culture plates, 6 M HCl (hydrochloric acid) was used to lower the pH of the A + B mixture to 6.5. The pH was measured using indicator strips. To prepare the filter paper mount for the embryo, 2 cm x 2 cm pieces of filter paper (Whatman) were punched with two adjacent 0.5 cm holes. Eggs were opened into a petri dish. The thick albumen was swept aside gently near the top of the yolk around the embryo. The filter paper was then lowered with the embryo showing through the punched holes and pressed onto the VM. The VM was then cut around the filter paper to release the embryo. The samples were further rinsed in PBS to remove excess yolk. The cleaned embryos were placed ventral side up on pre-warmed culture dishes. The embryo cultures were maintained in a slide box lined with wet paper towels in the egg incubators (37.5 °C ~45% humidity) except when snapshot imaging (<2 min per embryo) was performed. To perform the filter sandwich perturbation, an additional holed filter paper piece was attached to the ventral side of the embryo on the dish, and then wetted by PBS. To perform surgical cuts in the tissue to assess tension, embryos were first incubated for 4 h following the sandwich and control preparation described above. A sharp needle is then used to cut through the endoderm and ectoderm lateral to the body axis. After the surgery the embryos were imaged by an upright Kern microscope using recording mode with 10X magnification, focusing on the cut area.

## Double-ring culture and inflation system

The double-ring culture system was designed with modifications from a previous study[17]. The base and cover were printed using an in-house 3D printer (Ultimaker S5) using BASF Ultrafuse PLA (Inno3D). The Engineering Profile (0.15 mm) was used with 20% infill and no supports. A spherical cover glass (22 mm, VWR) was attached to the opening of the base ring by nail polish. The system is wetted with prewarmed culture medium (Part A only, see section above) before use. A 3 cm × 3 cm square of filter paper with a 2 cm diameter hole on the center was used to extract the VM and embryo from the egg. The filter culture embryo was then rinsed in PBS to remove excess yolk. Cleaned VM and embryo were placed on top of the cover glass ventral side up in the medium and clamped between the rings with screws. An externally attached syringe through a tunnel under the base ring allows culture medium to be supplied between the cover glass and the VM, inflating the clamped VM and achieving tension control. Inflation was controlled manually by slowly pressing the syringe to deliver ~1 ml of medium each time while observing the VM for rupture or leakage. The experiments were conducted on the stage of a stereomicroscope (10x objective, Kern Optics) and video recorded. The running time of each

experiment was usually short term (<10 min). With the exception of prewarmed medium, no other temperature control method was implemented.

## Equiaxial tension application

A stretcher device resembling a camera iris was designed and made to apply equiaxial tension to the embryo (See Supplementary Fig. 2f for a diagram and Supplemental files for a technical drawing of the device). The stretcher device consists of three major components: a base, a lid, and six blades. The base was a flat disc (diameter: 12 cm) with a circular hole (diameter: 4 cm) in the middle. In addition, a hexagonal trench was inscribed on the base disc to allow the blades to slide. The lid had the same dimension as the base, that is, a flat 12 cm disc with a 4 cm hole in the centre. The lid had six evenly spaced grooves arranged along the tangents to the central hole. The shape of the blade is derived from an equilateral triangle, with two angles being sliced off. Each blade had three additional components: two cylindrical studs facing upward and a cuboid protrusion facing downward. The stud at the tip of the blade was the site of attachment to the VM from the dorsal side of the embryo. The computerized models of these components were designed using Blender 3.0 (https://www.blender.org/). The.STL file of each component was then generated and 3D-printed using an in-house 3D printer (Ultimaker S5). When the device was assembled, the downward facing protrusion of each blade was placed into the hexagonal trench, whereas the upward facing cylindrical stud near the edge of the blade was inserted into the grooves in the lid. As such, when the lid is rotated, it drove the six blades to slide along the trench which caused the size of the central aperture to change. Since the VM is attached to the studs at the tip of each blade through small pieces of filter paper, as the aperture size enlarged, the VM was stretched and hence equiaxial tension was applied to the embryo. In each test, a HH8-12 embryo was cut from the yolk, transferred to the device and allowed to attach to the filter paper pieces glued to the studs. A thin layer of culture gel or albumen was then added to allow longer term culture of the embryo. The whole device was then transferred into a 15 cm petri dish with lid and then placed in the incubator.

## In ovo perturbations

Yolk extractions: a small window (~1.5 cm × 1.5 cm) was opened in the eggshell on D0 eggs with a scalpel blade and scissors. Subsequently a fine pipette was used to puncture the VM and extract yolk from the egg (~2.5 ml). For control embryos ~2.5 ml of yolk was removed and immediately reinjected. After yolk extraction the eggs were sealed with tape and incubated at 37.5 °C for 6 h (D0) or 38 h (D1.5). The embryos were extracted and imaged (Leica MZ10F, Zeiss Axio Observer 7). Samples that did not heal the VM resulting in yolk collapse would fail to develop embryos and are excluded from analysis. Oil treatment (adapted from[35]): Eggs were dipped in mineral oil (Sigma) and wiped clean of excess oil and returned to the incubators. The oil stayed on the eggshell pores stably over extended time (48 h maximum in this study) as observed under bright light. For heterochronic albumen transfer, a small window (~1.5 cm × 1.5 cm) was opened in the eggshell on D1 or D0 eggs and the albumen was extracted slowly using a pipette without damaging the yolk. Albumen from another day was then injected into the eggs. For the control samples the albumen was removed and reinjected. After the manipulation the eggs were sealed with tape and incubated at 37.5 °C for ~24 h to reach D2. The embryos were then extracted and imaged (Leica MZ10F).

## VM protein extraction and SDS-PAGE

~8 eggs per incubation time (0 h; 24 h; 36 h; 48 h; 54 h) were opened into petri dishes and the VMs were cut open, removed from the dish and washed in PBS. The obtained VMs were dried in a SpeedVac. Proteins were extracted from the samples by using a buffer consisting of 50 mM Tris−HCl (pH 8.0), 10% glycerol, 2% sodium dodecyl sulfate

(SDS), 25 mM ethylenediaminetetraacetic acid, and protease inhibitor cocktail (Sigma-Aldrich). Samples were incubated overnight under constant stirring at room temperature. After this, the samples were centrifuged (12000 $g$, 30 min, 4 °C), the supernatant was collected, and the concentration of proteins was determined using the Lowry method. Sodium dodecyl sulfate polyacrylamide gel electrophoresis (SDS-PAGE) was carried out using 1.0 mm thickness of both running gel (12%) and stacking gel (5%) in the presence of SDS. The electrophoresis was run at 8 V/cm in a Tris-HCl buffer system using a Vertical Biorad Mini attached to a Power Pac Basic electrophoresis apparatus. The samples were dissolved in 0.2 mol/L Tris-HCl buffer (pH 8.0) containing SDS (5 g/L) to a final concentration of 10 µg of protein/lane. Protein solutions (10 µl) along with a molecular weight ladder (260 kD, Thermo Scientific) were loaded onto the gel and stained with colloidal Coomassie blue (Colloidal Blue Stain Kit, Invitrogen) or with Pierce™ Glycoprotein Staining Kit (Thermo Scientific) after separation.

### Glycoprotein staining of the vitelline membrane

To examine the glycoprotein fibers on the VM, the samples were mounted on a cover slide and stained using Pierce™ Glycoprotein Staining Kit (Thermo Scientific), with PBS buffer wash to prevent drying. The stained VMs were examined microscopically at 100x (Kern Optics). To test the effect of pH changes on the VM glycoproteins, eggs without incubation (D0) were used to extract the VM. The samples were soaked in pH buffers of 7.5 and 9.3. The buffers were made with 0.1 M monopotassium phosphate and 0.05 M borax (pH 7.9 to 9.5) and 0.05 M phosphate buffer[61]. No adjustments were made for ionic strength. The VM samples were incubated overnight (12 h) at room temperature (~25 °C) or in the incubator (37.5 °C).

### Lysozyme-ovomucin interaction under different pH

Ovomucin was prepared as described[62]. Albumen was diluted with 5 volumes of 0.1 mol/L NaCl solution and stirred gently for 30 min using a magnetic stirrer. Subsequently, the pH of the mixture was adjusted to approximately 6 using 1 mol/L HCl. The dispersion was kept overnight at 4 °C and centrifuged at 15,000 $g$ for 20 min at 4 °C. The precipitate was suspended in 0.5 mol/L NaCl solution and then kept at 4 °C for 6 h. After another centrifugation at 15,000 $g$ for 20 min at 4 °C, the precipitate (containing ovomucin) was separated from the supernatant. The preparation was analyzed with SDS-PAGE to check for ovomucin. A few other protein bands were found to remain in the preparation but not expected to affect the following assay. The interaction between lysozyme and ovomucin was tested by turbidity as described[37]. Tubes containing the ovomucin preparation were diluted 1:2 with pH 7.0 to 9.3 buffers before the addition of purified lysozyme (2 ug/ml, Sigma). The contents were then mixed and incubated at 37 °C for 1 h. The absorbance was measured at 450 nm by a microplate reader (Perkin Elmer 1420).

### Sample preparation for electron microscopy

For the VM samples, eggs from different time incubation (0 h; 48 h; 54 h) were used for VM extraction. Samples were fixed in a fixative (2% glutaraldehyde/2% formaldehyde in 0.05 M sodium cacodylate buffer pH 7.4 containing 2 mM calcium chloride) overnight at 4 °C. After washing 5x with 0.05 M sodium cacodylate buffer at pH 7.4, samples were osmicated (1% osmium tetroxide, 1.5% potassium ferricyanide, 0.05 M sodium cacodylate buffer at pH 7.4) for 3 days at 4 °C. After washing 5x in DIW (deionised water), samples were treated with 0.1% (w/v) thiocarbohydrazide/DIW for 20 minutes at room temperature in the dark. After washing 5x in DIW, samples were osmicated a second time for 1 h at RT (2% osmium tetroxide/DIW). After washing 5x in DIW, samples were block-stained with uranyl acetate (2% uranyl acetate in 0.05 M maleate buffer at pH 5.5) for 3 days at 4 °C. Samples were washed 5x in DIW and then dehydrated in a graded series of ethanol (50%/70%/95%/100%/100% dry), 100% dry acetone and 100% dry acetonitrile, 3x in each for at least 5 min. Samples were infiltrated with

a 50/50 mixture of 100% dry acetonitrile/Quetol resin (without BDMA) overnight, followed by 3 days in 100% Quetol (without BDMA). Then, the sample was infiltrated for 5 days in 100% Quetol resin with BDMA, exchanging the resin each day. The Quetol resin mixture is: 12 g Quetol 651, 15.7 g NSA, 5.7 g MNA and 0.5 g BDMA (all from TAAB). Samples were placed in embedding moulds and cured at 60 °C for 3 days. For embryo samples, filter cultures were briefly rinsed with 0.9% saline and then fixed for 1 h at room temperature (2% formaldehyde/2% glutaraldehyde in 0.05 M sodium cacodylate buffer pH 7.4 containing 2 mM calcium chloride). Then, the long side of the filter paper window was gently removed on one side, leaving a C-shaped filter frame around the embryo. This was done to stabilize the thin embryo and keep it from collapsing, but also to minimize any potential distortion of the embryo due to shrinkage processes during the embedding. The embryo was then fixed in the above fixative for another 3 h at room temperature. The samples were washed 4x in 0.05 M sodium cacodylate buffer pH 7.4 and osmicated overnight at 4 °C (1% osmium tetroxide in 0.05 M sodium cacodylate buffer pH 7.4). Wash 5x in deionised water and dehydrate in a graded series of ethanol solutions (70%, 95%, 2 × 30 min in each) and infiltrate with a 50/50 mixture of LRW resin and 95% ethanol overnight at 4 °C. On the next day, exchange against 100% LRW resin and renew the resin 2x, incubating at RT during the day and at 4 °C overnight. Then, the embryos were embedded out in LRW - either in polyethylene cups or large PTFE moulds–covered with Aclar film to exclude air and the resin was cured at 60 °C overnight. Embryos were sectioned using a Leica Ultracut microtome and sections of about 120 nm thickness were placed on Melinex (TAAB) coverslips. The sections were post-stained in 2% uranyl acetate/50% methanol for 3 min and in Reynold's lead citrate for 6 min. After air drying, the coverslips were mounted on aluminium SEM stubs using conductive carbon tabs (TAAB) and sputter-coated with 30 nm carbon for conductivity using a Quorum Z150 TE carbon coater.

### TEM and SEM

For TEM (transmission electron microscopy) imaging of the VM, sections (70 nm thickness) were prepared in an ultramicrotome (Leica Ultracut) and placed on 300 mesh bare copper grids. Samples were viewed in a Tecnai G2 TEM (FEI/ThermoFisher) run at 200 keV accelerating voltage using a 20 µm objective aperture to improve contrast; images were acquired using an AMT digital camera. For SEM (scanning electron microscopy) imaging of the VM, thin sections were placed on Melinex (TAAB) plastic coverslips and allowed to air dry. Coverslips were mounted on aluminium SEM stubs using conductive carbon tabs (TAAB) and the edges of the slides were painted with conductive silver paint (TAAB). Then, samples were sputter coated with 30 nm carbon using a Quorum Q150 T E carbon coater. Samples were imaged in a Verios 460 scanning electron microscope (FEI/Thermofisher) at 4 keV accelerating voltage and 0.2 nA probe current in backscatter mode using the concentric backscatter detector (CBS) in field-free mode at a working distance of 3.5–4 mm; 1536 × 1024 pixel resolution, 3 µs dwell time, 4 line integrations. Stitched maps were acquired using FEI MAPS software (1.1.9.605) using the default stitching profile and 10% image overlap. For SEM imaging of the embryo sections, samples were also imaged in the Verios 460 scope at an accelerating voltage of 4 keV and a probe current of 0.2 nA using the CBS in field free mode for low magnification imaging and in immersion mode for high magnification imaging at a working distance of 2.5–4 mm; 1536 × 1024 pixel resolution, 3 µs dwell time, 4 line integrations.

### Data analysis

All data points are measured from distinct samples unless specifically described in the corresponding figure legends. Images and movies were processed by Fiji (NIH) and Powerpoint (Microsoft) for measurements which are further processed in Excel (Microsoft). Scale bars were first set with control images with objects of known sizes. For

**Article** https://doi.org/10.1038/s41467-023-38988-3

elongation speed, the distance between a fixed somite pair (usually somite 3 or 4) and the posterior end of the body axis was taken. For posterior neural tube (pNT) width and convergence speed, the distance between the outer boundary of the neural tube walls was taken. For glycoprotein fibre density measurements, the fibre images were first converted to binary images where the fibres are shown in white and the background shown in dark. Regions of interests were then randomly drawn on images to measure the intensities which were then averaged. For mechanical probe data, the raw trace (TDMS file, LabView) of each measurement containing the position of the probe and the force data over time was loaded into Matlab using custom code including a TDMS converter (Brad Humphreys, https://github.com/humphreysb/ConvertTDMS). The traces were examined alongside their corresponding live movies to exclude situations of early VM rupture and when probes hit the holder structure. The force curve of the admitted traces was first adjusted with their own baseline before probe movement started. The high-frequency data points were then averaged to one value per second. The average rate of force increase of a 20 s duration over the approximately linear phase of force curve was used to measure the stiffness of the samples.

### Reporting summary

Further information on research design is available in the Nature Portfolio Reporting Summary linked to this article.

## Data availability

All data are included in the paper and Supplemental Information. Source data are provided with this paper.

## Code availability

Custom codes used for the mechanical probe and data processing are provided as a supplemental file.

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

## Acknowledgements
The authors thank J. Rees, D. Shah, G. Wu, A. Sossick, P. Williamson, A. Downie, N. Smith, R. Butler, C. Jiggins, G. Sheng and members of the Xiong lab for reagents, technical assistance, and comments. This study is supported by a Wellcome Trust/Royal Society Sir Henry Dale Fellowship (215439/Z/19/Z) and a UKRI EPSRC Frontier Research Guarantee grant (EP/X023761/1, initially approved as a starting grant by ERC) to F.X.

## Author contributions
D.K., A.W. and F.X. designed and performed the experiments, analyzed the data and wrote the manuscript. C.U.C., C.R.B. and E.T. contributed to the ring culture device; R.H.P., W.W. and Y.Y.S.H. contributed to the mechanical probe system. F.G. and K.H.M. contributed to electron microscopy, which was performed using the facilities of the Cambridge Advanced Imaging Centre (CAIC).

## Competing interests
The authors declare no competing interests.
