## [Peer Review File · Nature Communications]

Downregulation of Extraembryonic Tension Controls Body Axis Formation in Avian EmbryosREVIEWER COMMENTS

Reviewer #1 (Remarks to the Author):

Poultry science has established that ageing of chick eggs affects properties like pH of the yolk and albumin as well as composition and mechanical properties of the vitelline membrane (VM) and that this effect may be due to the loss of CO₂ from the egg (From and Matrone 1962, Kirunda and McKee 2000, Biladeau and Keener 2009, Zou et al 2021, many others). Since the work of NEW it has been established that the tension on the vitelline membrane is a major determinant in the success of the development of the chick embryo in vitro. This paper now confirms several of the findings of the loss of tension and weakening of the VM (fig1) during early development and elaborates by correlating changes in the mechanical properties of the VM with the successful progression of the early stages of chick development, especially body axis elongation (fig2&3) and midline convergence speed (fig 4). The authors confirm that preventing changes in VM mechanical properties by limiting gas exchange, likely but possibly not only CO₂ over the egg-shell, halts the normally occurring increase in albumin pH, prevents loss of structural changes in the fibre organisation of VM and mechanical changes of the VM (fig 5). In the last set of results, it is shown that treatments preventing weakening of the VM by pH manipulation results in an increasing occurrence of body axis defects.

The need for a stiff VM during early development will need further clarification/validation. It is implied that early development needs a stiffer VM, while later development needs a less stiff VM. The evidence (Fig. 2) for a requirement of a stiff VM during early development is less convincing. Taking out some of the yolk, resulting in the release of VM tension clearly impairs development. It is suggested that this is due to detrimental effects on epiboly during the early stages of development, but it is not evident from the data provided whether this is the case. Epiboly was not measured directly and there is no indication whether this is especially important during the first day of development covering primitive streak formation and extension. Also the data from Fig 6 D,E would suggest that the pH manipulation results mostly in defects during axis elongation. How does this result compare to the experiments of Spratt who reported that when embryos are removed from the VM and placed on a semi-solid agar do develop well to the 10-20 somite stages. Can this semi-solid agar provide enough traction for development to proceed?

Overall, this is an interesting piece of work that combines a number of disconnected observations on age related structural changes in the mechanical properties of the VM and confirms these with newer mechanical measurements. Importantly it links these observations with a more detailed analysis of the consequences on early development, especially body axis elongation. How this comes about is as yet much less clear.

Minor points

In the albumin transfer experiment (Fig 6A) it is suggested that the effect is due to a change in pH of the albumin, which will then affect VM stiffness. This is possible, but it could also be the result of other changes in the chemical composition of the albumin. What happens if Day2 albumin (with high pH) is transferred to an incubated embryo?

In the protein blots (Fig S3B) the total amount of protein (D0) appears to be rather different from the D1 and D2 samples even though supposedly an equivalent amount of protein is loaded in each lane. This makes the interpretation of the PAS staining rather difficult. This should be better quantified. The description of the VM indentation setup could be improved, by using a diagram rather than the current rather vague photograph which makes it difficult to see the different components. It would help to have a technical drawing of the iris stretcher, indicating movement and direction of applied force (radial/tangential?) rather than the rather wordy description in methods.

Reviewer #2 (Remarks to the Author):

In this manuscript, Kunz and colleagues address the role of the extraembryonic mechanical forces on the development of the body axis, using chicken embryo as an experimental model. They first measure the tension of the vitelline membrane and find a downregulation over early developmental stages. With several experimental approaches to modulate tension, they show that this downregulation is necessary for correct body elongation, convergence and neural tube closure. They subsequently analyze the composition of the vitelline membrane and find a downregulation of glycoproteins in the outer layer, correlating with the relaxed tension. Finally, they show that this recomposition of the vitelline membrane is caused by the modulation of the pH of the albumen based on CO₂ exchange over time.

Overall, this is an interesting work addressing how the extraembryonic environment controls embryo morphogenesis. The experiments are well designed and methods properly described. Results are well presented and discussed. However, I would like to raise several points that require completion or clarification:

- The authors mention that the variation in yolk volume is minimal (line 303), but it would be worth adding a quantification to this manuscript.
- What is the somite number in embryos showing body axis defects? This should be monitored for all experiments of VM tension modulation, as in Figure 6a.
- Regarding somite number, authors should mention and discuss Nelemans et al.'s work (doi: 10.1016/j.isci.2020.100976) showing that embryo stretching causes defects in cell clustering during somitogenesis. Could VM tension modulation as presented by authors produce such defects in somitogenesis?
- Does the experimental modulation of tension (Figures 2, 3, 4, S2) induce a modification of VM composition compared to physiological control?
- In the albumen transfer experiment (Figure 6a), does transfer affect VM composition? Is this effect reversible?
- Regarding tension measurement (Figure 1, S1): is the tension homogenous across the whole vitelline membrane? Especially along the D/V axis, given that the embryo forms on top of the yolk, there could be an asymmetrical tension of the VM. This point should be experimentally characterized at best, discussed at least.
- Similarly, in all experiments presented here, the tension applied is homogenous in all lateral directions, notably in the stretcher experiment Figure 4c. Yet, the physical rearrangements of the embryo follow elongation along the antero/posterior axis, and convergence along the medio/lateral axis. So is it expected that the VM tension distributes evenly around the blastoderm? Conversely, one could expect that the developing blastoderm pulls the VM and creates tension along the A/P axis. Could the tension downregulation be unevenly distributed and axis-dependent around the blastoderm?
- Could the authors speculate on the possible tension variation of the area opaca and the area pellucida over the course of development? Regarding this point, they could discuss Lee et al.'s work (doi: 10.1242/dev.200303) and others'.
- Could the authors clarify why the VM tension may no longer be important? (lines 317-319) This assumption should be correlated with the hypothesis that the VM tension does not vary after D2 – is it the case?
- Could the authors elaborate on how extraembryonic tension may affect cell division, in particular by providing relevant references? (lines 336-337)
- What is the CO₂ source leading to a pH variation in the albumen? Is the pH variation similar in unfertilized egg? Given that the yolk index does not depend on fertilization, does tension depend on it? Does pH depend on it?
- Could the authors elaborate on the water exchange between albumen and yolk over the course of development? (line 299) The yolk aspect (viscosity, color) is very different between fertilized and unfertilized eggs, already at early developmental stages. Does this reflect a difference in density? Does the yolk density directly impacts the VM tension?

Minor points:

- could the authors explicit the formula to calculate VM tension in the Methods section?
- what is the positive control in Figure S3B?
- the numbers of embryos showing the different types of phenotype (body axis shortening, neural tube closure defects, rupture, etc) are well described in the text. It would be nicer to have a homogenous presentation in all figures, for example bar charts as in Figure 6e and/or pie charts with percentages.
- Line 195: use « another approach » or « an alternative approach ».

REVIEWER COMMENTS

Reviewer #1 (Remarks to the Author):

Poultry science has established that ageing of chick eggs affects properties like pH of the yolk and albumin as well as composition and mechanical properties of the vitelline membrane (VM) and that this effect may be due to the loss of CO₂ from the egg (From and Matrone 1962, Kirunda and McKee 2000, Biladeau and Keener 2009, Zou et al 2021, many others). Since the work of NEW it has been established that the tension on the vitelline membrane is a major determinant in the success of the development of the chick embryo *in vitro*. This paper now confirms several of the findings of the loss of tension and weakening of the VM (fig1) during early development and elaborates by correlating changes in the mechanical properties of the VM with the successful progression of the early stages of chick development, especially body axis elongation (fig2&3) and midline convergence speed (fig 4). The authors confirm that preventing changes in VM mechanical properties by limiting gas exchange, likely but possibly not only CO₂ over the egg-shell, halts the normally occurring increase in albumin pH, prevents loss of structural changes in the fibre organisation of VM and mechanical changes of the VM (fig 5). In the last set of results, it is shown that treatments preventing weakening of the VM by pH manipulation results in an increasing occurrence of body axis defects.

The need for a stiff VM during early development will need further clarification/validation. It is implied that early development needs a stiffer VM, while later development needs a less stiff VM. The evidence (Fig. 2) for a requirement of a stiff VM during early development is less convincing.

Taking out some of the yolk, resulting in the release of VM tension clearly impairs development. It is suggested that this is due to detrimental effects on epiboly during the early stages of development, but it is not evident from the data provided whether this is the case. Epiboly was not measured directly and there is no indication whether this is especially important during the first day of development covering primitive streak formation and extension. Also the data from Fig 6 D,E would suggest that the pH manipulation results mostly in defects during axis elongation.

Thank you for the constructive comment. In the revision we strengthened the evidence in Fig.2 by performing additional experiments looking at the epiboly/blastoderm spreading processes in the yolk-extracted embryos directly.

Taking advantage of the transgenic GFP embryos (McGrew et al., 2008) available from University of Edinburgh, we imaged the early (D0) blastoderm with clear contrast at the blastoderm edge which enabled us to measure the epiboly movement (spreading) of the blastoderm in the yolk-extracted and -reinjecting embryos. We found that reducing tension by yolk extraction reduces blastoderm expansion within the 6 hours of extraction (**new Figs. 2A-B**), which indicates epiboly defects. This result is consistent with classic experiments that detach the blastoderm edge from the VM or ruffle the VM to relax tension (New 1955; Bellairs et. al, 1963). As the reviewer points out, our results are consistent with and confirm these previous studies that establish an early requirement of VM tension for normal development. Our new experiments now add to the idea that early VM tension provides

proper condition for the epiboly process. The tension-reduced embryos that show a severely shortened body (Figs. 2C-E) therefore likely arise due to reduced primitive streak extension that requires a properly sized blastoderm.

The pH manipulation experiments with Oil in Figs. 6D,E result in the maintenance (but not decrease, Fig. 5G) of the early stage pH (and early stage VM stiffness according to our hypothesis). Since early stage VM tension in these embryos is preserved (Fig. 5H), early blastoderm/epiboly defects are not expected. Body axis defects are expected as the preserved strong VM persists into D1 and D2 when it resists the morphogenetic movements required for body axis development.

In the revision we further attempted alternate approaches to test the early requirement of stiffer VM. Decreasing the pH by acid injection into the D0 albumen led to complications as the embryos failed to develop and the results were not interpretable. We also performed albumen transfer where D2 albumen was transferred to D0 eggs. D0 eggs turned out to be very sensitive to this experimental procedure as compared to D1 eggs, in that the control transfer (D0->D0) causes developmental delays and defects. However these control embryos still developed a body axis and could be harvested. Interestingly, in contrast, the D2->D0 eggs (experimental group) show an extremely fragile VM. Some yolks were broken already when the eggs were opened, others broke upon opening. The few samples where we were able to look at the embryo, development appeared significantly delayed and no discernible body axis was observed. We therefore were unable to retrieve enough embryos to obtain meaningful body axis measurement such as those in Fig. 6A. These results however, are consistent with the idea that a higher pH albumen environment transferred to D0 eggs accelerates the decay of the VM, which weakens the VM and impairs early blastoderm development.

Taken together, the literature and the new results presented here are consistent with the idea that a stiffer early VM provides the higher level of tension required for blastoderm expansion.

How does this result compare to the experiments of Spratt who reported that when embryos are removed from the VM and placed on a semi-solid agar do develop well to the 10-20 somite stages. Can this semi-solid agar provide enough traction for development to proceed?

We speculate that in Spratt's experiments the tissues were able to attach to agar of proper stiffness to restore the required tension (the gel recipe of Spratt is essentially still in use following later adaptations for in vitro culture including in this work, in the presence of VM. When the gel is too stiff, the tissues show tearing. Chapman et al., 2001; our own observations). If the early VM and tissue tension level is mimicked, then it would be expected that the embryos will be able to develop to advanced stages similar to those observed in our oil experiments with largely normal anterior morphology and number of segments, but with some posterior body axis defects if such tension is not downregulated. Indeed, from the camera lucida drawings (Spratt 1947a, 1947b, Spratt and Haas, 1960), the embryos cultured with the Spratt method do appear to show more variable posterior axis morphology and are overall shorter than embryos cultured with VM suggesting an effect on

axis elongation. Nonetheless, it cannot be excluded that such posterior defects arise from other reasons (such as higher resistance to elongation by the gel compared to the in vivo environment).

The way in which tissues re-establish tension on the semi-solid agar remains unclear. In the case of the VM, the inner surface is a mesh of fibres (Bellairs et. al, 1963; Stafford et. al., 1985). During the first few days of incubation, the edge cells of the blastoderm are attached to this surface and use it as a substratum for movement as they actively expand the blastoderm (New 1955; Downie 1976). In the case of the agar, it's possible that more cells in addition to the edge cells participate in contact and contraction. These ideas can be tested by live imaging and traction force microscopy during blastoderm expansion in future.

We added these further discussions in the revision. The related paragraph now reads: *“Another interesting culture condition that is VM-free is the soft gel culture originally reported by Spratt (Spratt, 1947), in these experiments, the explanted blastoderms can spread and form a largely normal body axis with a shorter and smaller posterior. How the tissues re-establish tension on the gel substrate remains unclear, as the tissue-substrate contact area is presumably expanded from just the blastoderm edge in the case of the VM. It is possible that more cells in addition to the edge cells participate in contact and contraction on the gel. These ideas can be tested by live imaging and traction force microscopy during blastoderm expansion in future.”*

Overall, this is an interesting piece of work that combines a number of disconnected observations on age related structural changes in the mechanical properties of the VM and confirms these with newer mechanical measurements. Importantly it links these observations with a more detailed analysis of the consequences on early development, especially body axis elongation. How this comes about is as yet much less clear.

Thank you for the positive evaluation. To explore further how the consequences on body axis elongation come about, in the revision we performed new experiments where we looked closely at tension-increased embryos during body axis elongation stages with electron microscopy (**new Fig. 3I-J**). Tissue widening and separation between tissue layers are observed. Interestingly, tissues like the neural tube and notochord respond to increased lateral tension by becoming flatter, while the presomitic mesoderm (PSM) appears to stretch and show a lower cell density. These results show that different tissues respond to tension differently in the way they deform and change cellular organization. The apparent density reduction of the PSM tissue for example, could negatively impact elongation (Benazeraf et al, 2010).

In the revision we discuss these scenarios of how VM tension might affect body axis elongation to provide the reader with more context. We are currently investigating the gene expression, cellular and morphological changes of the body axis tissues under increased tension to identify possible responsive and regulatory mechanisms and will report them in future work.

Related discussion on the mechanisms linking tension to body axis defects now reads: *“In later stages (D1-D2), The convergence of the body axis tissues is of particular interest given*

that it is an inward tissue flow and is the most strongly affected process under high tissue tension in our experiments. Convergent movement and folding of the neural plate may involve the non-neural ectoderm generating a pushing force at the tissue transition that aids neural fold formation (Moury and Schoenwolf, 1995). One possible explanation for the failure of neural tube closure in our experiments is that increased tension on the VM limits the movement of these ectodermal cells, preventing the bending of the epithelial sheet that is important for proper folding. Alternatively, the non-neural ectoderm maintained under a higher level of tension could resist the contractile forces generated by the neural plate cells through apical constriction and interkinetic nuclear migration (Nikolopoulou et al., 2017). This would reduce tissue deformation while increasing the load on the cell junctions across the epithelial sheet, making it more likely for tissue ruptures to occur. In either scenario, the requirement of proper tension for tissue convergence raises possible mechanisms with which the neural plate and the non-neural ectoderm actively regulate tissue tension in response to environmental changes. Such mechanisms may include junctional remodeling and cell divisions (for example, cells orient the division axis to help relax tension (Campinho et al., 2013)), which can be tested in future studies using our platform. The increased tension also has an effect in the mesenchymal tissue of the posterior PSM that is different from the widening of the epithelial tissues such as the neural tube and notochord. In this case the PSM appears enlarged with a decreased cell density. This changes could underlie the slowing of body axis elongation as PSM cell density is known to regulate elongation through cell motility and/or PSM mechanical properties (Bénazéraf et al., 2010; Mongera et al., 2018; Xiong et al., 2020). In the anterior body axis, we observed relatively less changes of tissue morphology and delays likely due to the tissues (e.g., somites, anterior neural tube) being stiffer. It is worth noting that in tension-increased perturbations a lower rate of segmentation is observed. This suggests that tension can interfere with the somite formation process, which is consistent with previous studies where somite deformation and fission were observed when the embryo was subjected to a much higher strain (Nelemans et al., 2020; Stern and Bellairs, 1984)."

Minor points

In the albumin transfer experiment (Fig 6A) it is suggested that the effect is due to a change in pH of the albumin, which will then affect VM stiffness. This is possible, but it could also be the result of other changes in the chemical composition of the albumin. What happens if Day2 albumin (with high pH) is transferred to an incubated embryo?

We agree that the experiment presented in Fig. 6A alone does not rule out other changes in the albumen as the environment of the intact egg is complex and undergoes many changes during incubation, which may affect body axis development through changing VM stiffness or other mechanisms (such as viscosity of the albumen, buoyancy of the yolk, etc). This result serves as an independent *in ovo* test of the possible role of pH that complements the *in vitro* experiments reported here (Fig. 5C-D, 6B-C) and by others (Fromm, Cotterill and Winter) where only pH was changed. These *in vitro* results suggest a dominant role of pH on VM structure.

The literature and our results overall support the following scenario leading from pH changes to VM stiffness: pH increases weaken the interactions between the large proteins that make up the VM and the thick albumen around the yolk, which thins (viscosity

reduction) the thick albumen and deprives materials for VM maintenance, leading to VM thinning.

We performed the reverse transfer as suggested by the reviewer where D2 albumen was transferred to D0 eggs. D0 eggs turned out to be very sensitive to this experimental procedure as compared to D1 eggs, in that the control transfer (D0->D0) causes developmental delays and defects. However these control embryos still developed a body axis and could be harvested. Interestingly, in contrast, the D2->D0 eggs (experimental group) show an extremely fragile VM. Some yolks were broken already when the eggs were opened, others broke upon opening. The few samples where we were able to look at the embryo, development appeared significantly delayed and not advanced to body axis stages. We therefore were unable to retrieve enough embryos to obtain a meaningful body axis measurement as those in Fig. 6A. These results however, suggest that a higher pH albumen environment transferred to D0 eggs accelerates the decay of VM, which is consistent with our hypothesis and results using other methods such as yolk extraction in Fig.2.

In in vitro cultures, transferring D2 albumen to an embryo does not have a major effect on development. It is common to use D1/D2 albumen to make in vitro culture media as it is thinner and easier to handle (Chapman et al., 2001). In these situations VM tension is presumably maintained via the filter paper or other devices (e.g., rings in New's culture) in the early stages so that embryos are not as sensitive to VM structural decay caused by pH increase as in the in vitro conditions. Relaxing of the VM (e.g., when detached from the filter paper in EC culture) still causes blastoderm expansion defects in these cases. These observations show that D2 albumen does not cause early defects or body axis development defects on its own.

The D2->D0 results described above are included in the revision, results section around Fig. 6A and in the **new Table 1**.

In the protein blots (Fig S3B) the total amount of protein (D0) appears to be rather different from the D1 and D2 samples even though supposedly an equivalent amount of protein is loaded in each lane. This makes the interpretation of the PAS staining rather difficult. This should be better quantified.

We used the Lowry method to measure protein sample optical density and normalized the concentration before loading the lanes, details now added to the figure legend. The total amount loaded was similar between D0 and D1, D2 but some larger proteins were markedly reduced in D1, D2 samples, giving the impression of a smaller total amount. The lysozymes (~14kDa bands at the bottom of the gel) appear to be the dominant contributor to the total concentration in this assay. These results are consistent with Damaziak et. al., 2020, Mann, 2008. The PAS staining was performed on a different gel with the same set of samples, suggesting the loss of glycoprotein is even more pronounced than overall proteins. This has been clarified in the text and the legend.

To better quantify glycoprotein fibres specifically, which underpins the VM mechanical properties, in the revision we quantified the PAS-stained fibre density (**new Fig. 5I**) across

the different types of VM experiments we performed. This analysis shows that the PAS stained fibre density decreases from D0 to D2, consistent with the protein blot results.

The description of the VM indentation setup could be improved, by using a diagram rather than the current rather vague photograph which makes it difficult to see the different components.

We now included a detailed diagram of the indentation setup as a supplemental panel (**new Fig. S1E**). The diagram offers two orthogonal views that complement the photograph view from a different angle.

It would help to have a technical drawing of the iris stretcher, indicating movement and direction of applied force (radial/tangential?) rather than the rather wordy description in methods.

We now included a 3D-rendering of the iris stretcher as a supplemental panel (**new Fig. S3F**) to illustrate to readers how it works. Detailed technical drawings are available as a **Supplemental File** as requested by the reviewer. The driver moves tangentially and the contact points move mostly radially.

The method description is simplified accordingly.

Reviewer #2 (Remarks to the Author):

In this manuscript, Kunz and colleagues address the role of the extraembryonic mechanical forces on the development of the body axis, using chicken embryo as an experimental model. They first measure the tension of the vitelline membrane and find a downregulation over early developmental stages. With several experimental approaches to modulate tension, they show that this downregulation is necessary for correct body elongation, convergence and neural tube closure. They subsequently analyze the composition of the vitelline membrane and find a downregulation of glycoproteins in the outer layer, correlating with the relaxed tension. Finally, they show that this recomposition of the vitelline membrane is caused by the modulation of the pH of the albumen based on CO₂ exchange over time.

Overall, this is an interesting work addressing how the extraembryonic environment controls embryo morphogenesis. The experiments are well designed and methods properly described. Results are well presented and discussed. However, I would like to raise several points that require completion or clarification:

Thank you for the positive evaluation and constructive comments.

- The authors mention that the variation in yolk volume is minimal (line 303), but it would be worth adding a quantification to this manuscript.

In the revision we measured yolk volume from D0 to D2 (**new Fig. S1A**). We did not find significant changes of yolk volume in these stages, consistent with previous studies.

- What is the somite number in embryos showing body axis defects? This should be monitored for all experiments of VM tension modulation, as in Figure 6a.

Thanks for this important suggestion. In the revision we added somite number and segmentation speed information. Segmentation speed is added to experiments where elongation is being perturbed / compared. In these experiments, the embryos used were within the range of 8-12 somites at the beginning of the time course. This is standard for elongation speed analysis as established by previous studies (e.g., Denans et al., 2015). Somite number information is added for in ovo experiments to ensure that comparisons of tissue sizes (e.g., pNT width) drawn are at equivalent developmental stages.

Specifically:

Segmentation speed is added to Fig. 3D, 4E, 6B (SW, stretcher, low pH plate, respectively). The results show that in all cases the perturbations also cause segmentation speed to slightly slow down. These results are consistent with the idea that increased tension could affect the segmentation process, as shown by others (Neleman et al., Stern and Bellairs, 1984, see the following point). For inflation perturbation (Fig. 3B) the duration of perturbation is too short for a segmentation speed analysis to apply.

Somite number data are added to Fig. 6F. For yolk extraction experiments (Fig.2) the stage of tension effect started too early before the formation of any somites so a segmentation analysis is not informative.

- Regarding somite number, authors should mention and discuss Nelemans et al.'s work (doi: 10.1016/j.isci.2020.100976) showing that embryo stretching causes defects in cell clustering during somitogenesis. Could VM tension modulation as presented by authors produce such defects in somitogenesis?

Nelemans et al., is a very interesting study producing somite fission through axial tension. The total somite number changes as a result. A key distinction to the normal segmentation process from the fission condition reported by Nelemans et al is that fission take place after an initially normal somite forms. This mirrors classical embryology studies where somites were observed to divide into smaller ones laterally when the axis is under compression / lacks strain (Stern and Bellairs, 1984). We have not observed similar phenomena in our experiments, likely due to the fact that we introduced uniaxial tension and a moderate strain (as compared to Nelemans et al, which introduces ~50% strain). The strain in our experiments is likely not sufficient to induce the somite fission phenomena.

These studies have now been discussed in the revision. The paragraph reads: *"In the anterior body axis, we observed relatively fewer changes in tissue morphology and delays, likely due to the tissues (e.g., somites, anterior neural tube) being stiffer. It is worth noting that in tension-increased perturbations a lower rate of segmentation is observed. This suggests that tension can interfere with the somite formation process, which is consistent*

with previous studies where somite deformation and fission were observed when the embryo was subjected to a much higher strain (Nelemans et al., 2020; Stern and Bellairs, 1984)."

- Does the experimental modulation of tension (Figures 2, 3, 4, S2) induce a modification of VM composition compared to physiological control?

This is an interesting point concerning the feedback between tension and VM composition. It is possible. We think it's unlikely in short-term tension perturbations from under one hour to several hours (such as inflation, stretching, sandwich), as significant chemical changes would take time. To test this, in the revision we measured the VM glycoprotein fibre concentrations in SW conditions after 6 hours in D0 and D2 samples [**new Fig.5I**]. Although glycofibre is only one aspect of the overall VM composition, the results show that its density is not distinguishable from controls, suggesting no drastic VM structural changes in the short term under tension.

Longer term conditions (such as Fig.2 the ruffled/relax VM in ovo) may alter the way VM proteins interact with each other and/or with the albumen, changing VM composition. These would mean interesting feedback interactions that play into the regulation of tension. We currently do not have a high throughput way of assessing VM composition changes under different conditions. This can be further explored in future with proteomics and more enhanced protocols of VM preparation/washing.

We have described the SW results and raised the possibility of mechanical modification of VM composition in the revision.

- In the albumen transfer experiment (Figure 6a), does transfer affect VM composition? Is this effect reversible?

We did not measure the VM composition in these samples as the transfer experiments are difficult to perform and we have a low throughput (~20% successful transfers) to accumulate sufficient materials for protein composition analysis. These transfer experiments are expected to affect VM composition which subsequently affects VM mechanical properties. To address this question with an alternative method with higher throughput, in the revision we performed experiments where the extracted VMs are subjected to co-incubation with the albumen from a different date (**new Fig.5I and new panels in Fig.S3C**). These results show that incubated albumen in later days can weaken earlier VMs, which is consistent with the expectation that in ovo transfers may also have the same effect. Interestingly, putting D1/D2 VMs back to D0 albumen does not reverse the loss of fibres (**new Fig.5I and new panels in Fig.S3C**), suggesting that the structural decay of the VM is not reversible.

We also performed transfer of D2 albumen to D0 eggs, which led to an extremely fragile VM. Some yolks were broken already when the eggs were opened, others broke upon opening. These results suggest that a higher pH albumen environment transferred to D0 eggs accelerates the decay of VM, likely altering its composition.

- Regarding tension measurement (Figure 1, S1): is the tension homogenous across the whole vitelline membrane? Especially along the D/V axis, given that the embryo forms on top of the yolk, there could be an asymmetrical tension of the VM. This point should be experimentally characterized at best, discussed at least.

This is certainly a very interesting question. Since the hypothesis of this work focuses on the effect of tension on the developing embryo, which occupies a very small area (a few mm) on the top of the yolk (~5cm) and connects to the VM in that location through cells at the blastoderm edge, we focused on tension measurements with VM samples cut around the embryo area and not the rest of the VM.

It is possible that different areas of the much larger VM have different tension during development, which can provide even more information/guidance to the developing embryo than a uniform tension. Evidence that's consistent with this possibility includes classical measurements using a vacuum capillary by Fromm (Fromm 1964) on different locations of the VM where the author reported heterogeneities of VM strengths associated with the proximity of the VM to the chalazae (which are located to the equator ends of the yolk while the blastoderm is on the north pole). Although the tension pattern remains unknown. It's also important to note that many in vitro culture studies (Bellairs et al., 1967; Chapman et al., 2001; Dugan et al., 1991; New, 1959; Schmitz et al., 2016; Sydow et al., 2017) only involve the VM near the embryo, and early development in these conditions is mostly normal, suggesting that the mechanical properties of the VM that are far away from the embryo do not play an essential role in early development.

Unfortunately our current experiment approaches do not yet enable us to map out the tension heterogeneity on the entire VM. Our tension and stiffness measurements reported here and those by others rely on large VM pieces and therefore are not able to preserve or detect smaller scale or directional heterogeneities. To experimentally characterize this, a type of embedded sensor on the VM would be needed. Recent studies in tissues have taken advantage of actomyosin levels on the cell cortex and chemical tension sensors on the cell membrane, or inferred tension through imaging cell morphology. However, for the non-cellular protein-based VM these approaches are not applicable.

We incorporated these points in the revised discussion, which now reads: *"Whether the spatial pattern of mechanical properties of the VM (Fromm, 1964) contributes to this process remains unknown. Our structural and mechanical analysis of the VM samples taken around the blastoderm reported here has not revealed spatial heterogeneity, suggesting that the dynamics of VM tension may be uniform around the embryo. Therefore, patterned regulation of tension within the blastoderm will be essential for morphogenetic movements other than the expansion while VM tension provide a global baseline."*

- Similarly, in all experiments presented here, the tension applied is homogenous in all lateral directions, notably in the stretcher experiment Figure 4c. Yet, the physical rearrangements of the embryo follow elongation along the antero/posterior axis, and convergence along the medio/lateral axis. So is it expected that the VM tension distributes evenly around the blastoderm? Conversely, one could expect that the developing

blastoderm pulls the VM and creates tension along the A/P axis. Could the tension downregulation be unevenly distributed and axis-dependent around the blastoderm?

Thanks for this very interesting question. This relates to a discussion point that we made – namely how does the VM tension profile interact with the embryonic tension profile to dynamically drive proper morphogenesis. Characterizing and perturbing both and investigating the consequences on body axis formation is a key direction we hope to drive our future work towards.

The way the tensions are distributed and regulated around the embryo and along different axes remains an open area of investigation. We speculate the tension from the VM to be rather homogenous as we and others have not observed structural or organizational bias of VM proteins relative to the embryonic axes. And, as a passive protein membrane structure, the VM is not expected to have the type of dynamic symmetry breaking of live tissues. However, as the reviewer pointed out, we have some preliminary evidence that the blastoderm edge cells, that are connected to the VM and responsible for transmitting tension, could have a polarized pattern of activity (or ECM regulation) depending on their location (on the future anterior vs. posterior). Moving closer to the embryo, there is evidence that a patterned contractile actomyosin ring that drives the morphological flow including streak elongation and convergence (Saadaoui et al., 2020). These would suggest that tension is heterogeneously distributed along different embryo axes. This also explains our observation that increasing tension homogeneously in all directions has a stronger effect on the convergence process (medial-lateral) – which might already be under a higher tension within the tissues.

We think that the VM tension (and its downregulation) may be evenly distributed (at least around the embryo), these changes serve as an environmental/background tension level – which then interacts with unevenly distributed tension generated by different tissues which are more patterned and polarized by developmental signals. This interaction permits and/or promotes morphogenesis in some tissues.

The discussion section has been revised to provide more details on these questions. The paragraph now reads: *“A recent study shows that the area opaca serves an induction role to the marginal zone (Lee et al., 2022). It is tempting to speculate that these interactions may also alter the tension in the posterior area opaca and area pellucida, such as reducing it to facilitate primitive streak formation. A spatial-temporal tension map at high resolution using novel access tools and sensors (Datta et al., 2020) for the VM and the blastoderm will help test these ideas in future. In later stages (D1-D2), The convergence of the body axis tissues is of particular interest given that it is an inward tissue flow and is the most strongly affected process under high tissue tension in our experiments. Convergent movement and folding of the neural plate may involve the non-neural ectoderm generating a pushing force at the tissue transition that aids neural fold formation (Moury and Schoenwolf, 1995). One possible explanation for the failure of neural tube closure in our experiments is that increased tension on the VM limits the movement of these ectodermal cells, preventing the bending of the epithelial sheet that is important for proper folding. Alternatively, the non-neural ectoderm maintained under a higher level of tension could resist the contractile forces generated by the neural plate cells through apical constriction and interkinetic nuclear migration*

(Nikolopoulou et al., 2017). This would reduce tissue deformation while increasing the load on the cell junctions across the epithelial sheet, making it more likely for tissue ruptures to occur."

- Could the authors speculate on the possible tension variation of the area opaca and the area pellucida over the course of development? Regarding this point, they could discuss Lee et al.'s work (doi: 10.1242/dev.200303) and others'.

Lee et. al., 2022 reported that the area opaca functions to induce the posterior identity of the marginal zone. They found that the area opaca may signal to the marginal zone by emitting secreting molecules such WNT and BMP family members. However, as soon as the primitive streak forms (HH 2), the inducing and polarising abilities of the area opaca are lost. It is possible that these signaling activities, in addition to specifying cell fates, also alter the tension at the posterior area of the blastoderm around the marginal zone, for example by decreasing it locally, to facilitate the cell movements that form the streak. We think this is a very interesting speculation and included this in the revised discussion.

The relevant paragraph now reads: *"A recent study shows that the area opaca serves an induction role to the marginal zone (Lee et al., 2022). It is tempting to speculate that these interactions may also alter the tension in the posterior area opaca and area pellucida, such as reducing it to facilitate primitive streak formation. A spatial-temporal tension map at high resolution using novel access tools and sensors (Datta et al., 2020) for the VM and the blastoderm will help test these ideas in future."*

- Could the authors clarify why the VM tension may no longer be important ? (lines 317-319) This assumption should be correlated with the hypothesis that the VM tension does not vary after D2 – is it the case?

This speculation is from experimental evidence of perturbations such as the sandwich experiment that shows further changes of tension do not strongly affect body axis morphogenesis after D2. In ovo, the blastoderm edge cells and the forming vasculature continue to expand and cover the yolk by moving on the VM which likely still requires some VM tension (Downie 1976), but this interacting area is now quite far from the embryo proper which has completed convergence movement. The VM tension may stay the same or further weaken after D2, but it appears that body morphogenesis is no longer under its influence.

This has been clarified in Discussion as follows: *"Furthermore, sandwich experiments in late D2 embryos in this study do not show major body axis defects, suggesting that VM tension may no longer be important for the posterior body axis after the posterior body axis convergence stages."*

- Could the authors elaborate on how extraembryonic tension may affect cell division, in particular by providing relevant references? (lines 336-337)

Recent studies in different systems (e.g., Campinho et al., 2013) suggest that when the epithelial or epithelial-like tissue is under tension, cell division is promoted to relax that

tension. We are curious whether something similar could be happening in the ectoderm and endoderm of tension-increased embryos in our study, and whether that implies an active tension downregulation mechanism from the tissue during unperturbed development. This is currently being investigated in the lab.

This has been elaborated in Discussion as follows: *“the requirement of proper tension for tissue convergence raises possible mechanisms with which the neural plate and the non-neural ectoderm actively regulate tissue tension in response to environmental changes. Such mechanisms may include junctional remodeling and cell divisions (for example, cells orient the division axis to help relax tension (Campinho et al., 2013)), which can be tested in future studies using our platform”*

- What is the CO₂ source leading to a pH variation in the albumen? Is the pH variation similar in unfertilized egg? Given that the yolk index does not depend on fertilization, does tension depend on it? Does pH depend on it?

The high CO₂ in the egg initially comes from the hen before egg laying, which is equal to the CO₂ content of the mother’s body (Mueller 1958). We did not detect a significant difference of tension or pH between fertilized and unfertilized eggs in D0-D2 samples.

We noted the CO₂ source in the revision as follows: *“Previous studies have shown that the most drastic chemical change in the egg environment during these stages is an increase in the pH of the albumen from ~7.5 upon egg laying (when the albumen has a similar CO₂ content to the hen’s body) to 9-10 in D2 (Fromm, 1967; Mueller, 1958)”*

- Could the authors elaborate on the water exchange between albumen and yolk over the course of development? (line 299)

Yes. This discussion now reads: *“In addition to the biochemical modification of the VM, water influx from the albumen to the yolk may contribute to tension regulation (Moran, 1936) as such influx may increase the volume of the yolk resulting in increased pressure on the VM leading to higher tension.”*

The yolk aspect (viscosity, color) is very different between fertilized and unfertilized eggs, already at early developmental stages. Does this reflect a difference in density? Does the yolk density directly impacts the VM tension?

The color differences in yolk usually come from diet elements for the hen. We acquired unfertilized eggs from a different source in this study hence the observed color difference likely come from different diets at these sources. Some eggs we received from the fertilized egg source were also unfertilized (~10%). These eggs have the same color as fertilized ones, their yolk indexes are similar to the fertilized counterparts during D0-D3.

Generally, given the small mass and limited metabolic activities during the early stages, the developing embryo’s impact on the yolk is minimal until the yolk sac starts to function (D4). The yolk viscosity and biochemistry are not significantly different between fertilized and unfertilized eggs in the early days (Scalzo et al., 1970; Meuer and Egbers, 1990; Meng et al.,

2019). Furthermore, yolk viscosity would not affect the yolk index measured at the stable state in this study but rather the time to reach the stable state.

Yolk density does not change significantly in D0-D4 (Meuer and Egbers, 1990). We now also confirm this by measuring yolk weight and volume (**new Fig. S1A**). Yolk volume has an impact on VM tension as we have shown by yolk extraction. Yolk density change is not expected to change VM tension *in ovo* as long as yolk volume stays the same and the yolk remains afloat in the albumen.

Minor points:

- could the authors explicit the formula to calculate VM tension in the Methods section?

The equations used have been added to the methods section describing the scale tension measurement as follows: $\gamma(\text{native}) = [m(\text{after_cut}) - m(\text{detached})]g/L$, where m is the weight measurement, g is the gravitational acceleration 9.8m/s^2 , L is the width of the VM samples (2cm in the reported experiments). Similarly, $\gamma(\text{max}) = [m(\text{breaking}) - m(\text{detached})]g/L$.

This info is added to the Methods section under “Scale measurement of VM tension”

- what is the positive control in Figure S3B?

The positive control is Horseradish Peroxidase (~44kDa) from the staining kit.

This info is added to the figure legend of S3B.

- the numbers of embryos showing the different types of phenotype (body axis shortening, neural tube closure defects, rupture, etc) are well described in the text. It would be nicer to have a homogenous presentation in all figures, for example bar charts as in Figure 6e and/or pie charts with percentages.

Thanks for this suggestion. In the revision we added a summary table to homogeneously present the numbers of categorical phenotypes (e.g. rupture) to provide the readers with ease of reading and comparison of these phenotypes. For quantitative measures (e.g., body axis length, convergence speeds), we used the scatter plots showing all data.

Please refer to **Table 1**.

- Line 195: use « another approach » or « an alternative approach ».

Corrected to “another approach” at the corresponding line.

REVIEWERS' COMMENTS

Reviewer #1 (Remarks to the Author):

The authors have made a good attempt at addressing and clarifying the majority of the issues raised in my review. The additional information provided (figs 2a,b and 3i,j), although not definitive, is useful as is the enhanced discussion and improved technical documentation of the new stretch device used.

Reviewer #2 (Remarks to the Author):

The authors have addressed all my questions and added substantial information to the manuscript. In particular, I found the additional discussion and references very interesting and complementary to the study.

I recommend publication provided the few minor points below are addressed:

- I appreciate that the data about segmentation speed is now explicit in the figure legend, but they seem to be missing in each panel, while the authors stated they have been added (Fig. 3D, 4E, 6B). I would expect the scatterplots with individual replicate values to be added in the corresponding figures.
- Given the slight decrease observed, could the authors also add elements of discussion regarding this point? Is it expected to have an interdependence/equivalence between segmentation speed and elongation speed?
- The statistical information of Figure 5I is clearly explained in the figure legend but the annotation of the graph is counterintuitive – please use another symbol than the star to annotate non-significant comparisons. In addition, a one-way ANOVA with multiple comparisons is more appropriate. Authors don't have to annotate all pairwise comparisons but could at least highlight the differences between D0, D1 and D2.

NCOMMS-22-30829B

REVIEWERS' COMMENTS

Reviewer #1 (Remarks to the Author):

The authors have made a good attempt at addressing and clarifying the majority of the issues raised in my review. The additional information provided (figs 2a,b and 3i,j), although not definitive, is useful as is the enhanced discussion and improved technical documentation of the new stretch device used.

Thank you very much for the constructive comments that helped us improve the manuscript.

Reviewer #2 (Remarks to the Author):

The authors have addressed all my questions and added substantial information to the manuscript. In particular, I found the additional discussion and references very interesting and complementary to the study.

Thank you very much for the constructive comments that helped us improve the manuscript.

I recommend publication provided the few minor points below are addressed:

- I appreciate that the data about segmentation speed is now explicit in the figure legend, but they seem to be missing in each panel, while the authors stated they have been added (Fig. 3D, 4E, 6B). I would expect the scatterplots with individual replicate values to be added in the corresponding figures.

The segmentation data are now added to the relevant panels (Figs. 3d-e, 4e and 6b-c) as scatterplots.

- Given the slight decrease observed, could the authors also add elements of discussion regarding this point? Is it expected to have an interdependence/equivalence between segmentation speed and elongation speed?

This is an interesting question. In some conditions (e.g., surgery of posterior axis), elongation and segmentation speeds are not associated (ablating posterior PSM slows down elongation but doesn't affect segmentation rate), in other conditions (e.g., FGF signaling perturbation), there may be changes to both (during FGF inhibition, elongation speed slows down quickly while segmentation is affected in the longer term). The expectation is that segmentation and elongation are generally spatially and mechanistically separate and there isn't interdependence. In the cases described in this study, the perturbations applied (tension, pH etc) are on the whole embryo and they may affect segmentation and elongation separately at the same time.

We added a further comment on the interdependence in Discussion which now reads:

"It is worth noting that in tension-increased perturbations a lower rate of somitogenesis is observed. This suggests that tension can interfere with the somite formation process, which is consistent with previous studies where somite deformation and fission were observed when the embryo was subjected to a much higher strain^{56,57}. The observed changes to somitogenesis and elongation are unlikely to be associated but as separate responses to the globally applied tension perturbations."

- The statistical information of Figure 5I is clearly explained in the figure legend but the annotation of the graph is counterintuitive – please use another symbol than the star to annotate non-significant comparisons. In addition, a one-way ANOVA with multiple comparisons is more appropriate. Authors don't have to annotate all pairwise comparisons but could at least highlight the differences between D0, D1 and D2.

This panel has been modified accordingly.